# Sugar transporter Slc37a2 regulates bone metabolism in mice via a tubular lysosomal network in osteoclasts

Pei Ying Ng[1,14], Amy B. P. Ribet [1,14], Qiang Guo [1,2,14], Benjamin H. Mullin[1,3], Jamie W. Y. Tan[1], Euphemie Landao-Bassonga[1], Sébastien Stephens[4], Kai Chen[1], Jinbo Yuan [1], Laila Abudulai [1,5,6], Maike Bollen[5,6], Edward T. T. T. Nguyen [1], Jasreen Kular[1], John M. Papadimitriou[7], Kent Søe [8,9], Rohan D. Teasdale[10], Jiake Xu [1], Robert G. Parton [11,12], Hiroshi Takayanagi [13] & Nathan J. Pavlos[1] ✉

Osteoclasts are giant bone-digesting cells that harbor specialized lysosome-related organelles termed secretory lysosomes (SLs). SLs store cathepsin K and serve as a membrane precursor to the ruffled border, the osteoclast's 'resorptive apparatus'. Yet, the molecular composition and spatiotemporal organization of SLs remains incompletely understood. Here, using organelle-resolution proteomics, we identify member a2 of the solute carrier 37 family (Slc37a2) as a SL sugar transporter. We demonstrate in mice that Slc37a2 localizes to the SL limiting membrane and that these organelles adopt a hitherto unnoticed but dynamic tubular network in living osteoclasts that is required for bone digestion. Accordingly, mice lacking Slc37a2 accrue high bone mass owing to uncoupled bone metabolism and disturbances in SL export of monosaccharide sugars, a prerequisite for SL delivery to the bone-lining osteoclast plasma membrane. Thus, Slc37a2 is a physiological component of the osteoclast's unique secretory organelle and a potential therapeutic target for metabolic bone diseases.

Osteoclasts are bone-digesting cells that play a central role in skeletal bone growth and metabolism and underscore pathologies such as osteoporosis and osteopetrosis[1-3]. These multinucleated giants degrade bone via the ruffled border, a villous-like plasma membrane domain circumscribed by an actin ring that serves as the osteoclast's unique 'resorptive apparatus'. Ruffled border formation involves the polarized fusion of specialized lysosomal-related organelles (LROs), termed secretory lysosomes (SLs), with the bone-apposed plasmalemma[4,5]. Unlike conventional lysosomes, which traditionally serve as intracellular depots for the degradation and recycling of endogenous and exogenous biomolecules[6], LROs are 'hybrid organelles' that share features of both late-endosomes and lysosomes

[1]School of Biomedical Sciences, The University of Western Australia, Nedlands, WA 6009, Australia. [2]Department of Orthopaedics, The Third Xiangya Hospital, Central South University, 410013 Hunan, China. [3]Department of Endocrinology & Diabetes, Sir Charles Gairdner Hospital, Nedlands, WA 6009, Australia. [4]School of Medicine and Dentistry, Griffith University, Southport, QLD 4222, Australia. [5]Centre for Microscopy, Analysis and Characterisation, The University of Western Australia, Nedlands, WA 6009, Australia. [6]School of Molecular Sciences, The University of Western Australia, Nedlands, WA 6009, Australia. [7]PathWest Laboratory Medicine WA, Nedlands, WA 6009, Australia. [8]Pathology Research Unit, Department of Clinical Research, University of Southern Denmark, Odense C 5000, Denmark. [9]Department of Pathology, Odense University Hospital, Odense C 5000, Denmark. [10]School of Biomedical Sciences, The University of Queensland, Brisbane, QLD 4072, Australia. [11]Institute for Molecular Bioscience, The University of Queensland, Brisbane, QLD 4072, Australia. [12]Centre for Microscopy and Microanalysis, The University of Queensland, Brisbane, QLD 4072, Australia. [13]Department of Immunology, Graduate School of Medicine and Faculty of Medicine, The University of Tokyo, Tokyo 113-0033, Japan. [14]These authors contributed equally: Pei Ying Ng, Amy B.P. Ribet, Qiang Guo. ✉e-mail: nathan.pavlos@uwa.edu.au

coupled with exocytic functions[7]. Osteoclast SLs store the acidic hydrolase cathepsin K[8] and tartrate-resistant acid phosphatase[9] and are compositionally and functionally defined by the presence of several endo-lysosomal membrane proteins, namely LAMP1/2[10], Rab7[11], the *a3* V-ATPase proton pump subunit[12], chloride channel ClC-7[13], TI-VAMP/VAMP7 and synaptotagmin 7[14]. Fusion of SLs with the bone-apposed plasmalemma discharges cathepsin K into the underlying resorptive microenvironment to digest collagenous (Type 1a) bone matrix. At the same time, the coalescence of SL membranes with the ventral plasmalemma enriches the ruffled border with nanoscale bone-digesting machinery, including V-ATPases[15] and ClC-7[13], which cooperatively acidify the resorptive space to dissolve bone mineral. The ruffled border, together with the juxtaposed resorption lacuna, is therefore considered a giant digestive 'extracellular LRO'[16].

As a precursor membrane for ruffled border genesis, SLs represent fertile grounds for the discovery of new homeostatic regulators of bone mass and thus potential anti-resorptive drug targets. Yet despite their critical importance, our understanding of the molecular anatomy and spatiotemporal organization of osteoclast SLs remains limited. Here, to expand the molecular landscape of osteoclast SLs, we systematically surveyed the proteome of enriched SLs isolated from murine osteoclasts and identified Slc37a2 as a candidate SL transporter. Unexpectedly, using live cell imaging, we reveal that Slc37a2+ SLs exist as a hitherto unreported but highly dynamic tubular lysosomal network in osteoclasts that radiates throughout the cytoplasm and fuses with the bone-lining plasmalemma. Moreover, deletion of *Slc37a2* in mice results in a high bone mass phenotype attributed to osteoclast dysfunction and associated disturbances in SL resolution and delivery to the ruffled border. Altogether, our findings unmask Slc37a2 as an SL sugar transporter critical for bone metabolism and highlight previously unappreciated plasticity of the osteoclast's specialized lysosome-related organelle(s).

## Results

### Proteomics of osteoclast SLs identifies Slc37a2 as a candidate regulator of bone mass

To survey the proteome of osteoclast SLs, we adapted two well-established lysosome enrichment protocols using superparamagnetic iron oxide nanoparticles (SPIONs)[17,18]. For this, large-scale murine bone marrow monocyte (BMM)-derived pre-osteoclast cultures (Day-3, post-RANKL stimulation) were 'pulsed' for 24 h with SPIONs to encourage uptake into endosomes. As a derivative of endomembranes, we hypothesized that endocytosed nanoparticles could be 'chased' into SLs upon the convergence of SPION-loaded endosomes with lysosomes and secretory pathways, a process that is synchronized with the final stages of RANKL-mediated differentiation of pre-osteoclasts into mature $\alpha\nu\beta_3$-integrin-positive (IntegriSense[645]) osteoclasts (>90% of cells) (Fig. 1a, b). Following a 24 h 'chase' (total 'pulse-chase' = 48 h), osteoclasts were homogenized, SPION-loaded organelles captured from post-nuclear supernatants (PNS) using magnetic columns, and membranes serially eluted in fractions (F1–F3) (Fig. 1c). The enrichment of isolated SPION-loaded organelles in the Fraction F2 preparation was confirmed by transmission electron microscopy (TEM) (Fig. 1c) and their origins assessed immunoanalytically (Fig. 1d). Gratifyingly, captured F2-organelles showed substantial enrichment with major SL membrane markers including LAMP2 and Rab7, as well as the primary osteoclast luminal acidic hydrolase cathepsin K (Ctsk) and the LRO resident GTPase Rab38[19] but were virtually devoid of contaminating organelles (Fig. 1d), thus deeming them suitable for proteomics.

The protein composition of SL-enriched F2 fractions was analyzed by high-resolution mass spectrometry (nLC-ESI-MS/MS). A representative mass spectrum of fraction F2 is depicted in Fig. 1c (far right panel). In total, 4153 different proteins were unambiguously identified and defined by at least two unique peptides and master proteins. Of

these, 218 achieved >1.5 Log₂fold change (FC) and adjusted $P < 0.05$ relative enrichment in SL F2-fraction compared to the starting cell homogenate by label-free quantitation (Fig. 1e). The results are summarized in Supplementary Data 1. As expected, the SL proteome was enriched with membrane proteins, including major SL resident proteins ClC-7 (Clcn7, enriched 2.35 Log₂FC, $P = 3.51E^{-02}$) and V-ATPase 116 kDa *a3* subunit (Tcirg1, enriched 2.01 Log₂FC, $P = 1.66E^{-03}$), thus validating our approach. In addition, the SL proteome was enriched with proteins involved in trafficking ($n = 14$) including Rab38 (enriched 2.62 Log₂FC, $P = 3.56E^{-03}$) as well as core subunits of the mammalian guanidine exchange factor for Rab7 i.e. Mon1b (enriched 2.66 Log₂FC, $P = 7.63E^{-03}$)[20] and Regulator of Mon1-Ccz1 complex (Rmc1/*C18orf8*; enriched 3.06 Log₂FC, $P = 2.72E^{-03}$)[21] (Fig. 1e). Intriguingly, the SL proteome was dominated by proteins involved in molecular transport ($n = 43$), with almost half representing members of the solute carrier (Slc) protein superfamily of secondary active transporters ($n = 16$)[22] (Supplementary Data 1). Finally, several miscellaneous proteins were also identified, some of which constitute major cargo of endolysosomes, for example, β-hexosaminidase (enriched 1.74 Log₂FC, $P = 4.05E^{-03}$) and others that are predicted to localize to endolysosomal membranes but whose SL residency remains to be confirmed (e.g. CD68, enriched 2.25 Log₂FC, $P = 1.95E^{-02}$) (Fig. 1e).

To first gauge the relevance of the mouse osteoclast SL proteome to humans, we investigated whether significant associations identified in a large human genome-wide association study (GWAS) for estimated bone mineral density (eBMD)[23] were enriched among human homologs of the mouse osteoclast SL proteome gene set. The top 218 proteins enriched in the SL F2-fraction are mapped onto a CIRCOS plot, with gene groupings representing those with human homologs strongly associated ($P < 6.6E^{-9}$, green) and very strongly associated ($P < 5E^{-20}$, red) with estimated BMD indicated (Fig. 2a). Using the VEGAS2Pathway approach[24], we confirmed statistically significant enrichment of eBMD association signals in the SL gene set ($P$ EMPIRICAL = 0.04), indicating that the SL proteome is conserved in humans and that it is enriched with candidates clinically-relevant to the regulation of human bone mass.

To further validate the SL proteome, we next focused our attention on membrane transport proteins. We chose this protein family for two reasons: (i) they are polytopic proteins that contain multiple transmembrane domains (TMDs) and are thus integral components of the SL limiting membrane and; (ii) mutations in genes encoding SL transporters underscore most (>70%) of the known forms of human autosomal recessive osteopetrosis (ARO)[25]. To this end, the SL proteome was further filtered for 'Membrane Transport Proteins' that: (i) possessed >1 TMD ($n = 42$); ii) housed a lysosomal targeting signal(s) outside of their TMDs (i.e. YxxØ or [DE]xxxL[LI], where Ø stands for an amino acid residue with a bulky hydrophobic side chain) ($n = 22$); and; (iii) whose corresponding lead SNPs reached strong genome-wide significance in the human eBMD dataset ($P < 6.6E^{-9}$). Six candidates satisfied these criteria: (i) Tcirg1 and (ii) Atp6v0a1, which encode the mammalian 116 kDa *a3* and *a1* subunits of the osteoclast vacuolar proton pump, respectively with mutations in *a3* alone accounting for over 50% of infantile malignant ARO[26]; (iii) Clcn7, a chloride voltage-gated channel indispensable for osteoclast function in mice and in humans[13]; (iv) Atp2b1, a P-type Ca²⁺-ATPase that regulates bone mass by fine-tuning osteoclast differentiation and survival[27]; (v) Slc17a5, a lysosomal H⁺-driven sialic acid transporter whose mutations underpin Salla disease, a lysosomal storage disorder accompanied by bone malformations[28] and; lastly (vi) Slc37a2, a little-studied member of the Slc37 family (isoforms a1–a4)[29] that although was not the most enriched protein detected on SLs, was enriched 2.35 Log₂FC ($P = 3.11E^{-02}$) and whose human homolog harbors a genome-wide significant signal for eBMD (lead SNP rs7949048, $P = 3.1E^{-12}$) (Fig. 2a, b, red box), thus prompting our interest and further investigation.

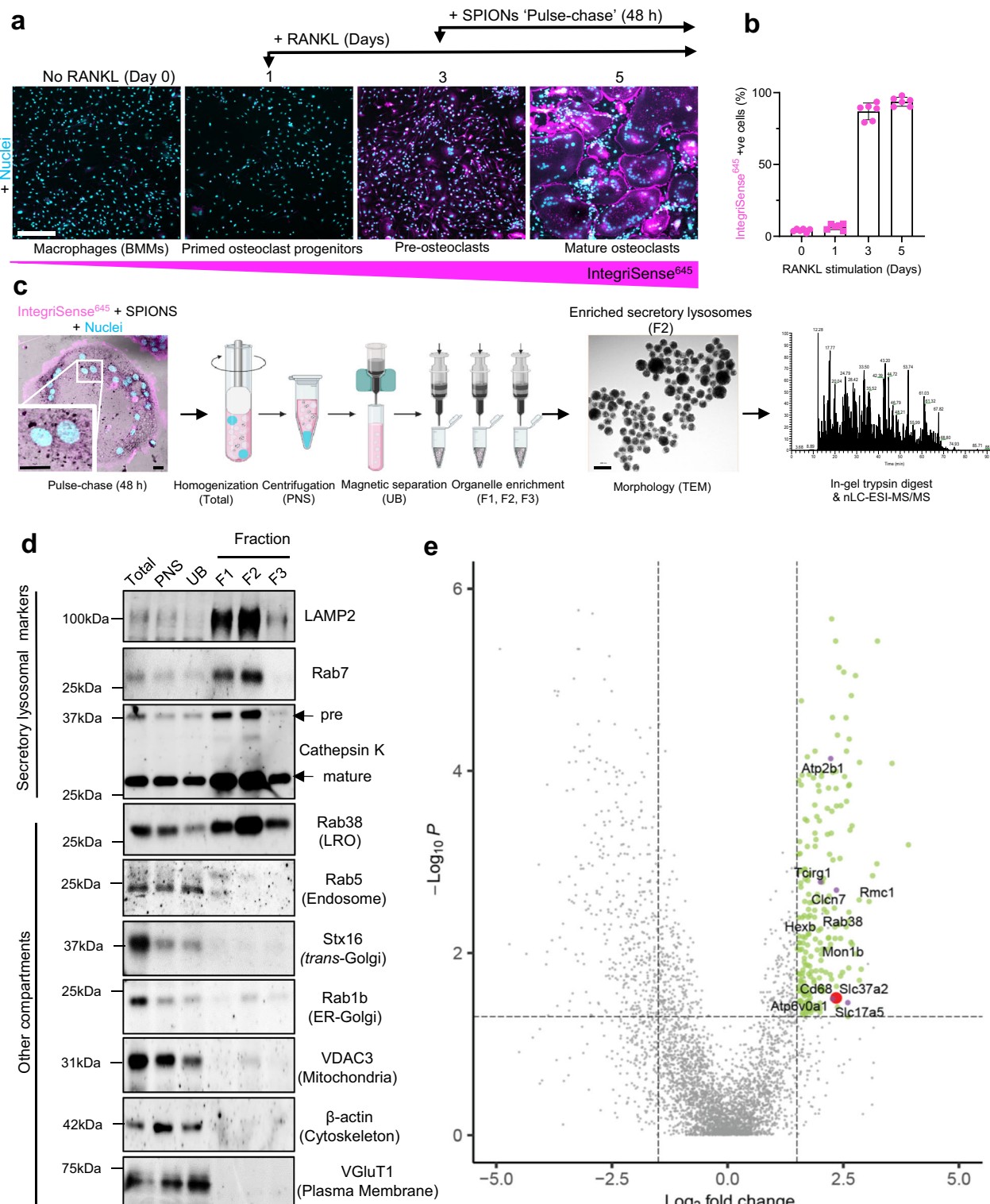

**Fig. 1 | Proteomic analysis of enriched osteoclast secretory lysosomes identifies Slc37a2. a** Experimental setup for SPION 'pulse-chase' and representative confocal images of mouse bone marrow monocyte (BMMs) differentiation into mature αvβ3-integrin-positive (IntegriSense645) osteoclasts following RANKL stimulation. Bar 150 μm. **b** Quantitation of IntegriSense645 positive cells during RANKL stimulation (*n* = 3 biologically independent cells from 2 experiments). Data are presented as mean percentage (%) ± SD. **c** Schema of the SPION-based SL enrichment method, validation, and proteomic analysis. Created with BioRender.com.

**d** Immunoblotting for protein markers of various subcellular compartments in whole-cell homogenates (H), post-nuclear supernatant (PNS), magnetic column flow-through (UB), and enriched SL fractions (F1–F3) (*n* = 3). LRO lysosome-related organelle. **e** Volcano plot depicting the difference between proteins in whole cell homogenates and secretory lysosomes (SLs) (*n* = 3, *P* > 0.05, ANOVA, Benjamini–Hochberg adjusted). The top 218 proteins up-regulated in the lysosomal fractions are shown in green (FC > 1.5, *P* < 0.05) with the top 6 membrane transporters colored in purple and Slc37a2 highlighted in red.

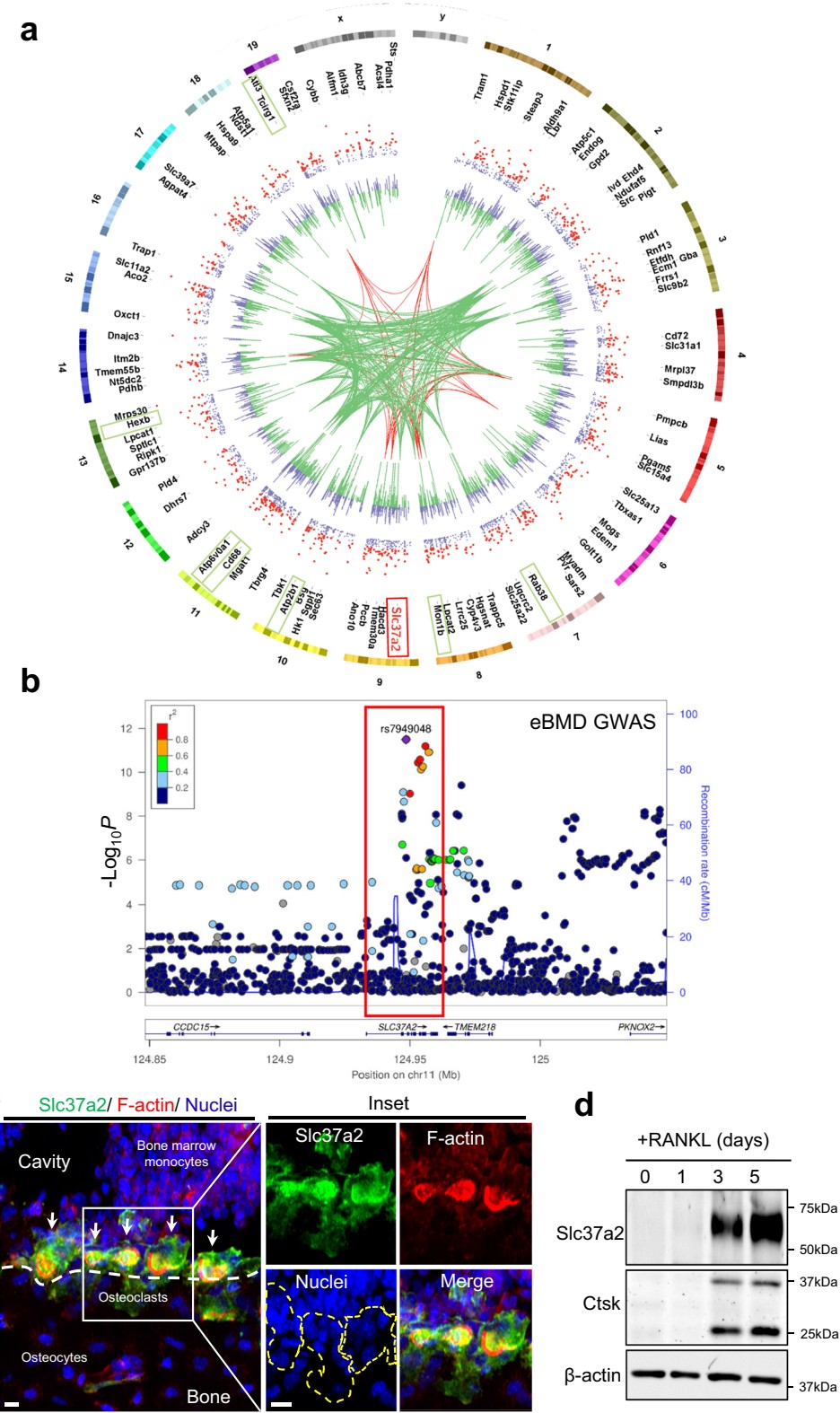

**a**

**b**

**c**

**d**

## Slc37a2 is expressed in mature osteoclasts and localizes to a network of tubular SLs

To explore the relevance of Slc37a2 to osteoclasts and bone biology, we first assessed Slc37a2 expression in long bones using specific antibodies raised against mouse Slc37a2 (Supplementary Fig. 1). As shown in Fig. 2c, Slc37a2 is highly expressed in mature osteoclasts, here lining endocortical bone, with immunofluorescent signal concentrated within F-actin rings as well as diffusely labeling the cell surface. In comparison, Slc37a2 expression is below the detection limit in neighboring bone marrow, osteoblasts, and osteocytes. Consistent with this expression pattern, Slc37a2 protein levels were robustly increased during RANKL-induced differentiation of BMMs into osteoclasts (Fig. 2d). Thus, these findings imply that osteoclasts are the major, if not exclusive, source of Slc37a2 among bone-lineage cells.

**Fig. 2 | Slc37a2 is a candidate regulator of human bone mass and is highly expressed in mature osteoclasts. a** CIRCOS plot[71] displaying mouse chromosomes, gene names for the top 218 proteins up-regulated in the SL fractions, scatterplot representing protein abundance ratio (secretory lysososomes (SLs)/homogenate) adjusted *P* values (purple) with significant associations (*P* < 0.05) colored red, histogram representing Log$_2$FC abundance values for each protein SLs/homogenate purple (+ve), green (−ve), gene groupings representing human homologs associated with eBMD at *P* < 6.6E$^{-9}$ (green) and *P* < 5E$^{-20}$ (red)[23]. Protein abundance ratio adjusted *P* values (ANOVA, Benjamini−Hochberg) are displayed as −Log$_{10}$ *P* values. Red box denotes Slc37a2 and green boxes other notable genes. **b** Regional association plots for the human *SLC37A2* gene (red box) generated using estimated bone mineral density (eBMD) GWAS association results[23]. Genetic variants within 100 kb of the lead variant (rs7949048, purple) are depicted (*x*-axis) along with their eBMD *P* value (−Log$_{10}$) generated using a linear mixed non-infinitesimal model (LMM). **c** Slc37a2 expression in the femur of 12-week-old male WT mice by immunohistochemistry. Arrows indicate osteoclasts and a dashed line defines bone (white) and osteoclast (yellow) surfaces. Bars, 10 μm (*n* = 3). **d** Slc37a2 expression dur*in*g in vitro osteoclast differentiation of bone marrow monocytes (BMMs) by immunoblotting. Cathepsin K (Ctsk) and β-actin served as controls (*n* = 3). Source data are provided as a Source Data file.

Next, we examined the subcellular distribution of endogenous Slc37a2 in osteoclasts. First, we used immunoblotting to validate independently the enrichment of Slc37a2 during the biochemical isolation of SLs from mouse osteoclasts (Supplementary Fig. 2a). To verify its endogenous localization in intact osteoclasts, we immunostained cultures of mouse BMM-derived osteoclasts using the same Slc37a2-specific antibody along with a panel of organelle identity markers against endo-lysosomes (LAMP2, Rab7, Arl8), early/recycling endosomes (Vps35), endoplasmic reticulum (ER, protein disulfide isomerase, PDI), and the Golgi (GM130). As shown in Fig. 3a, Slc37a2 labeling resulted in a tubulo-vesicular staining pattern that closely overlapped with endo-lysosomal proteins LAMP2, Rab7, and Arl8 but not endosomes, the ER or Golgi as confirmed by quantitative co-localization analyses (Fig. 3b). Closer inspection revealed that whereas Rab7 colocalized with large Slc37a2$^+$ puncta (Fig. 3a, yellow arrows) Slc37a2$^+$ tubular structures were decorated primarily with Arl8 (Fig. 3a, blue arrows) in keeping with the known overlapping roles of these two GTPases in lysosome positioning and tubulation[30].

Among the Slc37 family, Slc37a2 is unique, existing as an N-terminal glycosylated 12-transmembrane spanning protein (Fig. 3c) encoded by two naturally occurring splice variants (isoforms 1 and 2) that differ by five amino acids found within their extreme C-terminus (Fig. 3c, magenta box). Slc37a2 isoform 2 harbors a canonical lysosomal sorting signal (YxxØ) whereas isoform 1 possesses an extended alternative sequence SSxxxLTH-x that is conserved both in humans and in mice (Fig. 3d). Therefore, to differentiate the subcellular distribution of the two Slc37a2 isoforms in osteoclasts, the N-terminus of each mouse Slc37a2 variant was tagged with either emGFP (isoform 2) or mCherry (isoform 1) and cloned into a CMV-driven mammalian vector. Because osteoclasts are refractory to conventional transfection methods, tagged-Slc37a2 variants were co-expressed by direct nuclear microinjection into mouse BMM-derived osteoclasts. Consistent with its endogenous distribution, when microinjected into live osteoclasts and imaged by time-lapse confocal microscopy on glass, Slc37a2 isoforms co-occupied an expansive network of tubulo-vesicular compartments that were highly dynamic and radiated throughout the osteoclast cytoplasm (Fig. 3e–h, Supplementary Movie 1). As shown in Fig. 3e, whereas the subcellular distribution of $^{emGFP-}$Slc37a2 isoform 2 tightly overlapped with $^{mCherry-}$Slc37a2 isoform 1 on tubular organelles, Slc37a2 isoform 1 showed an additional preference for the plasma membrane (PM, magenta outline), likely reflecting its alternative C-terminal targeting signal. The lumens of these tubular organelles were acidic and housed cathepsins/K, as confirmed by the accumulation of the acidophilic probe LysoTracker Red (Fig. 3f) and cathepsin/k fluorescent substrates DQ-BSA (pan-cathepsins) (Fig. 3g) and Magic Red (cathepsin K MR) (Fig. 3h), indicating that they were of endolysosomal origin. This was further corroborated by monitoring co-trafficking of $^{emGFP-}$Slc37a2$^+$ organelles with $^{mRFP-}$LAMP1 as well as LRO marker proteins $^{mCherry-}$VAMP7 and $^{mCherry-}$Rab38 in living osteoclasts (Supplementary Fig. 2b, c, Supplementary Movie 1). By comparison, while Slc37a2$^+$ tubules closely intersected with other organelles including early endosomes ($^{mCherry-}$Rab5) and mitochondria (Mitotracker Red), they were morphologically and dynamically distinct (Supplementary Fig. 2b, c and Supplementary Movie 2). The endo-

lysosomal nature of $^{emGFP-}$Slc37a2$^+$ compartments was also verified in non-osteoclastic cells (Supplementary Fig. 3).

The osteoclast Slc37a2$^+$ tubular lysosomal network was highly sensitive to chemical and cold fixation (i.e. 4% paraformaldehyde, Supplementary Fig. 4a) and rapidly collapsed upon treatment with the destabilizing reagent nocodazole (Fig. 3i), indicating that it is intimately linked to microtubules[31]. In addition, Slc37a2$^+$ tubules frequently projected towards and made contact with the osteoclast plasma membrane (Supplementary Movie 3). When cultured on devitalized bone discs, $^{emGFP-}$Slc37a2$^+$ tubules concentrated within F-actin rings demarcating ruffled borders (Fig. 4a, b) and fused with the bone-lining plasma membrane when monitored under live settings (Fig. 4c–f, Supplementary Movie 4). While the bone disc physically obstructed direct visualization of fusion between $^{emGFP-}$Slc37a2$^+$ tubules and the ruffled border membrane, a transient fusion between tubules and the osteoclast plasma membrane (PM) was captured at the level of the bone surface (Fig. 4e, f). Here, $^{emGFP-}$Slc37a2$^+$ tubules extended towards and made prolonged contact with the bone-lining PM (Fig. 4e, yellow line) before transiently fusing, as monitored by content release (Fig. 4f, white arrows and line scans), and then rapidly retracting, a process reminiscent of tubular lysosome-to-PM fusion observed in dendritic cells[32]. Importantly, these tubular compartments were not an artifact of $^{emGFP-}$Slc37a2 expression, as the same network of acidified, cathepsin-containing tubular organelles could be visualized in naïve osteoclasts using vital probes LysoTracker and/or DQ-BSA by optical (Supplementary Fig. 4b, c, Supplementary Movie 5) and by electron (Supplementary Fig. 4d) microscopy. Collectively, these data indicate that Slc37a2 occupies a network of acidified cathepsin-containing tubular LROs that fuse with the bone-lining plasma membrane, thus fulfilling the usual criteria of osteoclast SLs. Herein, we refer to Slc37a2$^+$ compartments as 'tubular SLs'.

### Slc37a2 deficiency leads to high bone mass

To investigate the physiologic importance of the Slc37a2$^+$ tubular SL network in osteoclasts and bone metabolism, we next generated Slc37a2 knockout (KO) mice using targeted embryonic stem cells obtained from the 'Knockout Mouse Project' consortium. Genotyping and qPCR analysis confirmed the correct insertion of the tma2 targeting cassette and that the approach produced an effective knockdown of the *Slc37a2* mRNA (>95% reduction in *Slc37a2*) in bones from homozygous mice compared to wild-type (WT) littermates (Supplementary Fig. 5a–c). *Slc37a2$^{tm2a(KOMP)wtsi}$* homozygous mice, referred herein as "*Slc37a2* knockout" (*Slc37a2$^{KO}$*) mice, showed no overt differences in the size or weight of matched littermates up to 24-weeks-of-age (Supplementary Fig. 5d). Similarly, no obvious abnormality in skeletal patterning was observed in whole mount alcian blue/alizarin red staining of skeletal preparations of 5-day-old *Slc37a2$^{KO}$* mice (Supplementary Fig. 5e) or by whole body X-ray of 12-week-old female mice (Supplementary Fig. 5f). On the other hand, radiographs and microcomputed tomography (μCT) analysis of the distal femurs (Fig. 5a–d) of 12-week-old WT and *Slc37a2$^{KO}$* mice revealed that the long bones of null mice were radiodense and exhibited a dramatic increase in trabecular bone volume; reaching an impressive ~3–10-fold

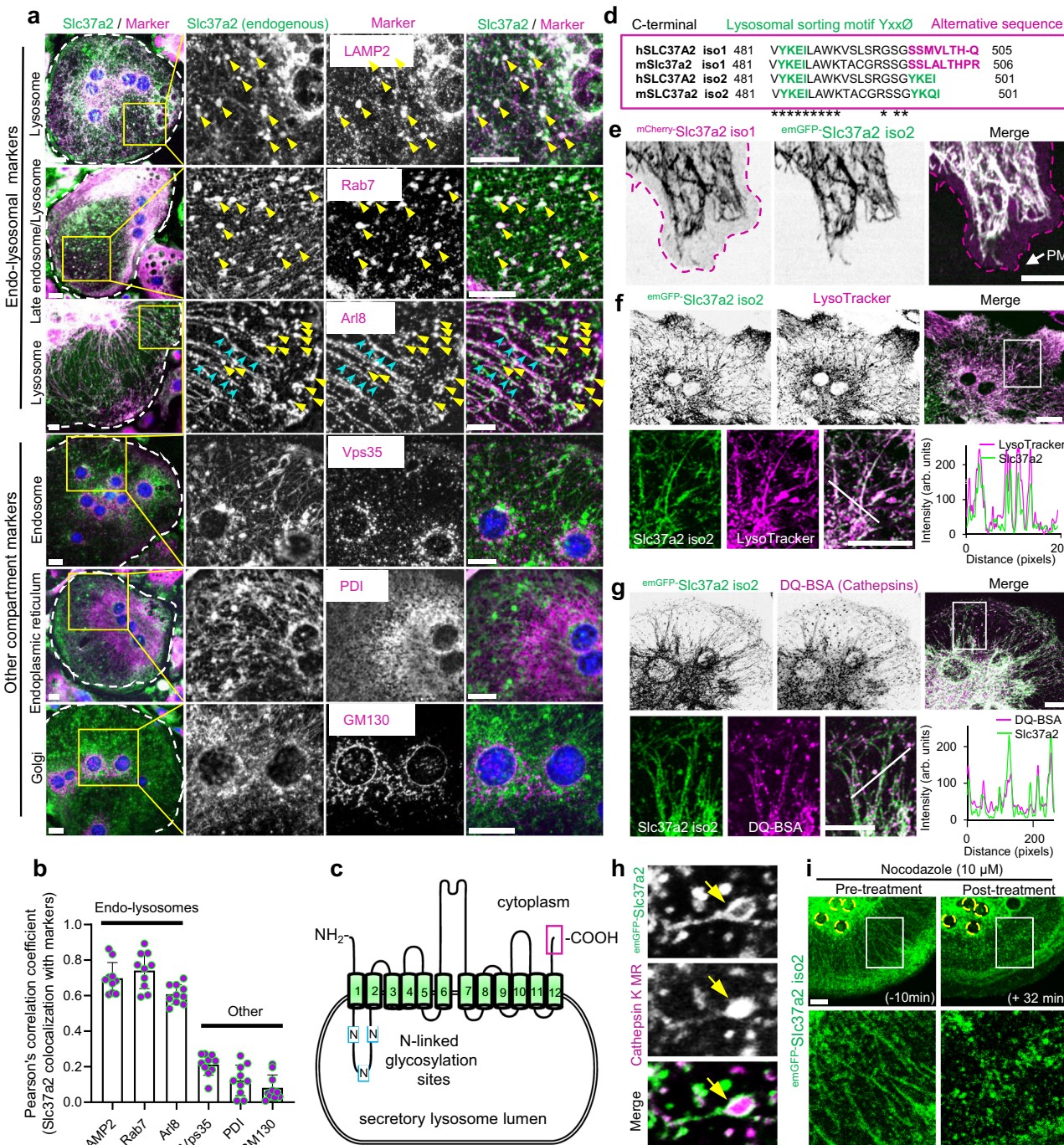

**Fig. 3 | Slc37a2 localizes to a network of tubular secretory lysosomes.**
**a** Endogenous localization of Slc37a2 in mouse bone marrow monocyte (BMM)-derived osteoclasts. Slc37a2 co-localizes with late endo-lysosomal markers (LAMP2, Rab7, and Arl8), but not with endosomes (Vps35), the endoplasmic reticulum (PDI) or the Golgi (GM130). Bar, 10 μm. **b** Pearson's correlation coefficient (Rr) calculated from 10 osteoclasts pooled from two independent experiments. Data are presented as means ± SD. **c** Schematic of Slc37a2 structure and topology on SL membranes, with transmembrane (green cylinders), luminal cytosolic domains (black lines), N-linked glycosylation sites (blue boxes, N), and extreme C-terminus (magenta box) indicated. **d** Amino acid alignment of the far C-terminus for human SLC37A2 and mouse Slc37a2 isoforms (1 and 2) with lysosomal sorting motifs and alternative sequences indicated. Asterisks indicate conserved amino acids. **e** Confocal image of a mouse osteoclast cultured on glass co-microinjected with ^mCherry-Slc37a2 isoform 1

and ^emGFP-Slc37a2 isoform 2. Purple dashed line outlines the plasma membrane (PM). Bar, 10 μm (n = 5). **f** and **g** Representative confocal image of a live mouse osteoclast on glass microinjected with ^emGFP-Slc37a2 isoform 2 and pulsed with endolysosomal probes LysoTracker Red (**f**, n = 3) or DQ-BSA (**g**, n = 3). Line scans of the individual fluorescent intensities (arbitrary units, arb. units) correspond to the white diagonal line in the magnified views. Bars, 10 μm. **h** High-resolution live confocal image of a representative tubular SL bearing ^emGFP-Slc37a2 isoform 2 and housing Cathepsin K Magic Red (MR) within its lumen (yellow arrow). Bar, 2 μm. (n = 3). **i** Time-lapse confocal image of an osteoclast expressing ^emGFP-Slc37a2 isoform 2 before (−10 min) and after (+32 min) treatment with the microtubule disrupting agent nocodazole (10 μM). Bar, 10 μm (n = 3). Source data are available in the Source Data file. See also related Supplementary Movies 1–3.

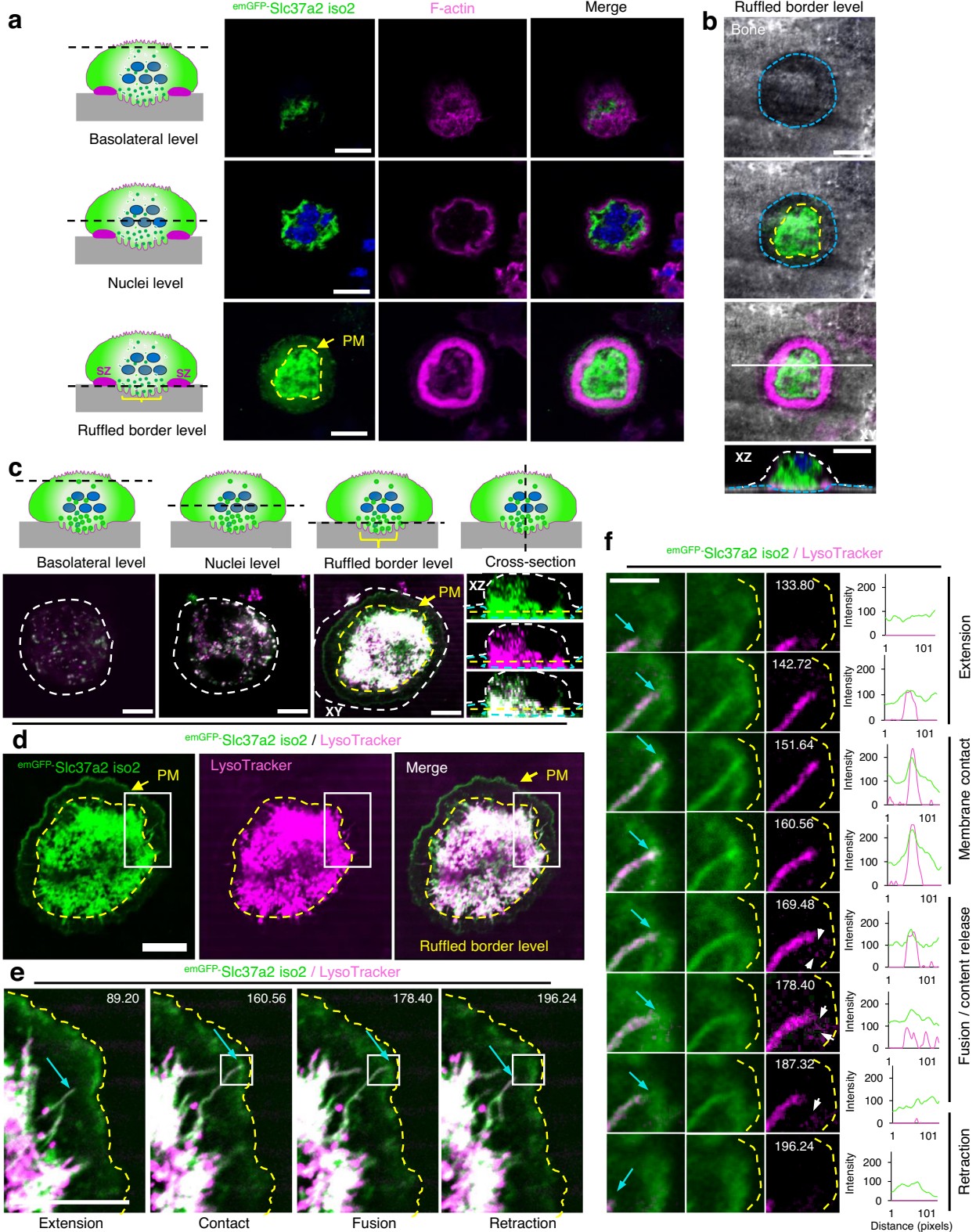

increase in BV/TV in male and female mice, respectively (Fig. 5e). Consistent with this finding, femurs of *Slc37a2*^KO mice had increased trabeculae and a decrease in trabecular separation (Fig. 5f, g), but only females showed a significant increase in trabecular thickness at 12-weeks (Fig. 5h). Cortical bone thickness was also significantly increased in both male and female mice at 12-weeks (Fig. 5d and i) which imparted greater bone stiffness and whole bone strength as confirmed by three-point-bending tests (Fig. 5j, k). Elevated bone mass was also

evident in heterozygous (*Slc37a2*^HET) mice, indicative of a gene dose effect (Supplementary Fig. 5g, j–n). This profound increase in bone mass extended to vertebrae within the axial skeleton (Fig. 5l–n) as well as to the bones of the skull, albeit to a lesser extent (Supplementary Fig. 5h). While femur length was unaltered (Supplementary Fig. 5i), trabecular bone mass accumulated and was retained with age (Fig. 5o–q). These gains in bone mass conferred protection against ovariectomy-induced trabecular bone loss in vertebrae, and partially in

**Fig. 4 | ᵉᵐᴳᶠᴾˉSlc37a2⁺ tubules orientate towards the ruffled border and fuse with the bone-apposed plasma membrane. a** and **b** Multilevel confocal images of a mouse osteoclast cultured on bone expressing ᵉᵐᴳᶠᴾˉSlc37a2 isoform 2 and stained with rhodamine-phalloidin to indicate the F-actin ring and underlying ruffled border. Representative XY serial confocal sections depict ᵉᵐᴳᶠᴾˉSlc37a2 distribution at the basolateral, nuclear (blue), and ruffled border level(s) (n = 3). XZ denotes the side view corresponding to the white horizontal line in the merged panel. Dashed lines indicate ruffled border regions within sealing zones (yellow), resorptive pit (blue) and cell border (white). Bars 10 μm. **c** Multilevel XY and XZ confocal images of a live osteoclast cultured on bone expressing ᵉᵐᴳᶠᴾˉSlc37a2 isoform 2 and labeled with LysoTracker Red. Dashed lines denote the plasma membrane (PM, white),

sealing zone (yellow), and bone surface (blue). Bars, 10 μm. (n = 3). **d** Bottom-up view of the ruffled border level with the sealing zone denoted by the yellow dashed line. Bar, 10 μm. (n = 3). **e** and **f** Magnified regions of a time-lapse series illustrating extension, membrane contact, transient fusion, and retraction of an ᵉᵐᴳᶠᴾˉSlc37a2⁺ tubule (blue arrow) with the bone-lining plasma membrane (yellow dashed line), with time points, indicated. White arrows highlight content release/exocytosis indicative of tubule fusion with the PM as confirmed by corresponding signal intensities in fluorescence line scans intensities (arbitrary units). Bar, 2 μm. (n = 2). Osteoclast cartoons depicted in panels **a** and **c** are adapted from ref. [72]. See also related Supplementary Movies 4, 5.

femurs, but not against femoral cortical bone loss (Supplementary Fig. 6).

Histologically, *Slc37a2*ᴷᴼ femurs contained cartilaginous bars deep within trabeculae of the metaphysis (Fig. 6a, inset), suggestive of impaired calcified cartilage degradation during endochondral bone growth. Modeling of *Slc37a2*ᴷᴼ mice bones was also abnormal as evidenced by the broadening of the distal diaphysis and metaphysis (Fig. 6b, dual-arrows). Moreover, femurs of *Slc37a2*ᴷᴼ mice exhibited conspicuous fibro-cartilaginous lesions that occurred at the level of the periosteum and extended along the proximal metaphyseal–diaphyseal junction (Fig. 6c–h). This lesion stained intensely for TRAP activity, a marker of osteoclasts (Fig. 6e) and was interspersed with small TRAP⁺αᵥβ₃⁺ cells that lined the periosteal bone surface as well as numerous bone/cartilage fragments (Fig. 6e, f, arrows) that appeared partially demineralized under transmission electron microscopy (Fig. 6g, arrows). In addition, this periosteal lesion was densely populated with Runx2⁺ mesenchymal cells that stained strongly for MMP13, a matrix metalloprotease known to participate in the digestion of cartilage and collagen matrix (Fig. 6h). Similar lesions were observed along tibiae but not vertebrae of *Slc37a2*ᴷᴼ mice (Supplementary Fig. 7a). Outside of the skeleton, macroscopic and histopathological evaluation of major organs and tissues revealed no obvious abnormalities in *Slc37a2*ᴷᴼ mice at 12 weeks (Supplementary Fig. 5o, p). Taken together, these data suggest that the major phenotype of mice with global *Slc37a2* insufficiency is high bone mass.

## High bone mass in *Slc37a2*ᴷᴼ mice is driven by impaired bone resorption and compensatory increases in remodeling-based bone formation

To determine the cellular mechanism(s) by which *Slc37a2*-deficiency increased trabecular bone volume, histomorphometric and fluorescence immunohistochemical analysis of the trabecular surfaces of the femur were performed. First, osteoclast parameters were assessed and revealed an increase in both the total number of TRAP⁺ osteoclasts (-246%, P < 0.0001) and the number of osteoclasts lining trabecular bone surfaces (-182%, P < 0.0001) of *Slc37a2*ᴷᴼ mice by histomorphometry (Fig. 7a–c). This increase in osteoclast numbers was corroborated by elevated systemic levels of TRAP5b (Fig. 7d) as well as increased expression of key osteoclast marker genes Acp5 and Ctsk in femurs (Fig. 7p). Reflecting this, RANKL mRNA was robustly amplified in *Slc37a2*ᴷᴼ femurs, whereas OPG and RANK mRNA levels were unchanged, leading to a significant increase in the mRNA ratio of RANKL- to-OPG (Fig. 7e). Because enhanced osteoclastogenesis can result from hypocalcaemia that stimulates PTH release which, in turn, induces RANKL expression[1], we also checked serum calcium and PTH levels in *Slc37a2*ᴷᴼ mice but found no difference compared with WT controls (Supplementary Fig. 7b). In situ assessment of trabecular surfaces revealed that whereas *Slc37a2*ᴷᴼ osteoclasts were capable of forming F-actin rings, they were larger than WTs (Fig. 7f–h). Despite this, serum C-terminal telopeptides of type I collagen (CTX-I), a by-product of collagen type-1a that is released from bone into the blood by osteoclast resorption, was not significantly altered in *Slc37a2*ᴷᴼ mice

(Fig. 7i). However, the ratio between CTX-I and TRAP5b, which is used as an index for resorption per osteoclast[33], revealed a highly significant decrease in resorption per osteoclast (P = 0.0006) (Fig. 7j). Intriguingly, resorptive lacunae underlying αᵥβ₃⁺ *Slc37a2*ᴷᴼ osteoclasts were frequently occupied by flattened bone-lining MMP13⁺ cells (Fig. 7k, arrows), reminiscent of 'reversal' cells[34]. Along with the increased expression of MMP13, the mRNA levels of MMP9 and MMP14 were also significantly elevated in *Slc37a2*ᴷᴼ whole femurs (Fig. 7p).

Because osteoclast-mediated bone resorption is intimately coupled to the bone formation by osteoblasts[35], we also examined bone formation parameters by dynamic bone histomorphometry of double calcein-labeled frozen sections (Fig. 7l). As shown in Fig. 7n, femora isolated from *Slc37a2*ᴷᴼ mice exhibited significantly increased mineral apposition rates (MARs) compared with WT. Consistently, the number of osteoblasts occupying bone surfaces (Fig. 7m, o), as well as the expression of osteoblast marker genes Runx2 and ALP (Fig. 7p), were increased in distal femur metaphysis of 12-week-old *Slc37a2*ᴷᴼ mice. The capacity of *Slc37a2*ᴷᴼ osteoblasts to differentiate and form bone mineralizing nodules in vitro, however, was indistinguishable from WT osteoblasts (Supplementary Fig. 7c), implying that the observed increase in bone formation in vivo was indirect. Corroborating this, analyses of the mRNA expression of key osteoclast-derived 'clastokines' (Sphk1/S1P, Sema4D, Wnt10b, Wnt16, and BMP6) and the osteoblast 'coupling factor' (TGF-β)[35] in whole femurs revealed that loss of Slc37a2 in mice corresponded with a significant increase in Sphk1 and TGF-β1 mRNA expression, but not Sem4A (P = 0.06) Wnt16, Wnt10b or BMP6 (Fig. 7p). The mRNA levels of Sphk1 and TGF-β1 in WT and *Slc37a2*ᴷᴼ osteoclasts cultured in vitro were however comparable (Supplementary Fig. 7d), indicating that their elevated in vivo expression was a consequence of increased osteoclast numbers, and/or increased expression in other bone cells. Thus, high bone mass in *Slc37a2*ᴷᴼ mice is driven, primarily, by diminished bone resorption by osteoclasts together with imbalanced remodeling-based bone formation by osteoblasts.

## *Slc37a2* deletion impairs the function but not the formation of osteoclasts

We next assessed whether Slc37a2 regulates cell-autonomous effects in osteoclasts. To this end, BMMs isolated from WT and *Slc37a2*ᴷᴼ littermates were exposed to M-CSF and RANKL for 5 days to induce osteoclast differentiation. As shown in Fig. 8a, b, the number of TRAP⁺osteoclasts formed in vitro is indistinguishable between WT and *Slc37a2*ᴷᴼ mice. Consistently, no obvious morphological distinctions were observed in either the spreading or capacity of *Slc37a2*ᴷᴼ osteoclasts to form F-actin rich podosome-belts (Fig. 8c). Temporal expression of protein markers for osteoclast differentiation, including NFATc1, c-Src, V-ATPase or cathepsins K and B, were also comparable between WT and *Slc37a2*ᴷᴼ cells (Fig. 8d). Further, absence of *Slc37a2* did not alter key RANKL-induced signaling events, including phosphorylation of ERK1/2, Akt, and IκBα (Fig. 8e). Thus, Slc37a2 does not regulate osteoclast differentiation.

In comparison, when cultured on synthetic bone mineral (Fig. 8f) or bovine bone slices (Fig. 8g), *Slc37a2*ᴷᴼ osteoclasts

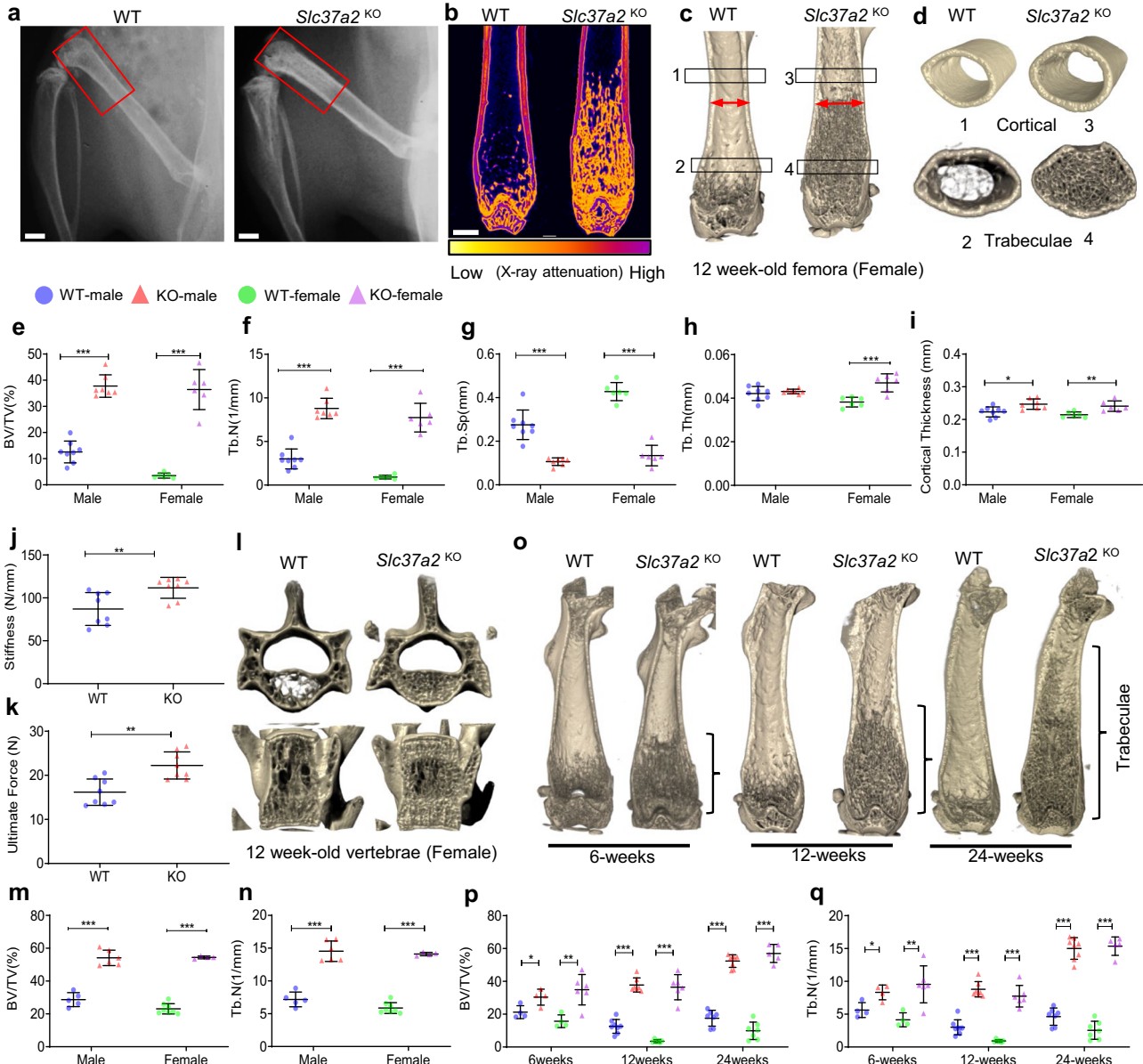

**Fig. 5 | *Slc37a2*KO exhibits high bone mass and increased bone strength.**
**a** Radiography of the hindlimbs of 12-week-old female WT and *Slc37a2*KO mice. Bar, 1 mm. **b**–**d** Representative sagittal μCT section false-colored to indicate X-ray attenuation (**b**) and μCT 3D reconstructed images of sagittal (**c**) and transverse (**d**) sections of distal femurs of 12-week-old female WT and *Slc37a2*KO mice. **e**–**i** μCT analysis of femurs of 12-week-old male and female WT (*n* = 8 male and 6 female) and *Slc37a2*KO (*n* = 7 male and 6 female) mice. **e** Male *P* < 0.0001 and female *P* < 0.0001. **f** Male *P* < 0.0001 and female *P* < 0.0001. **g** Male *P* < 0.0001 and female *P* < 0.0001. **h** Male *P* = 0.7874, female *P* < 0.0001. **i** Male *P* = 0.0114, female *P* = 0.0053. **j** and **k** Biomechanical three-point bending test of 12-week-old male WT and *Slc37a2*KO femurs. Stiffness (**j**, *P* = 0.0083) and ultimate force (**k**, *P* = 0.0013) are presented (*n* = 8 mice per genotype). **l**–**n** Representative μCT images of vertebrae (**l**) and μCT analysis of vertebrae (**m** and **n**) of 12-week-old male and female WT (*n* = 5 male and 8

female) and *Slc37a2*KO (*n* = 6 male and 4 female) mice. **m** Male *P* < 0.0001 and female *P* < 0.0001, **n** male *P* < 0.0001 and female *P* < 0.0001. **o**–**q** Representative μCT images of female femurs (**o**) and μCT analysis of femurs (**p** and **q**) of 6-, 12- and 24-week-old male and female WT (6-weeks *n* = 4 male and 4 female; 12-weeks *n* = 8 male and 6 female; 24-weeks *n* = 7 male and 7 female) and *Slc37a2*KO (6-weeks *n* = 4 male and 6 female; 12-weeks *n* = 7 male and 6 female; 24-weeks *n* = 8 male and 5 female) mice. **p** 6-weeks male *P* = 0.026112, female *P* = 0.004736; 12-weeks male *P* < 0.0001, female *P* < 0.0001; 24-weeks male *P* < 0.0001, female *P* < 0.0001. **q** 6-weeks male *P* = 0.014847, female *P* = 0.006968; 12-weeks male *P* < 0.0001, female *P* < 0.0001; 24-weeks male *P* < 0.0001, female *P* < 0.0001. All data are presented as means ± SD. **P* < 0.05, ***P* < 0.01, ****P* < 0.001, by two-tailed unpaired Student's *t*-test. Source data are available in the Source Data file.

exhibited a dramatically diminished capacity to acidify their extracellular environment and digest mineralized matrix. Unlike the well-demarcated resorption lacunae excavated by WTs, resorption pits generated by *Slc37a2*KO osteoclasts were morphologically irregular, formed fewer 'trench-like' resorptive trails, and were visibly shallow as revealed by reflective and scanning electron microscopy (Fig. 8g–k). Confirming this, the projection of confocal z-stacks of representative resorptive pits revealed the shallower temporal

depth and reduced collagen Type 1a exposure in pits formed by *Slc37a2*KO osteoclasts compared to WTs (Fig. 8l), reflecting their reduced extracellular acidification function. Intriguingly, this resorption deficit was independent of their capacity to form F-actin rings/sealing zones (Fig. 8m). Moreover, the pit phenotype could be rescued by transduction of recombinant lentivirus containing emGFP-Slc37a2 into osteoclasts derived from *Slc37a2*KO BMMs (Fig. 8n, o), confirming that the impairment was Slc37a2-dependent. Together,

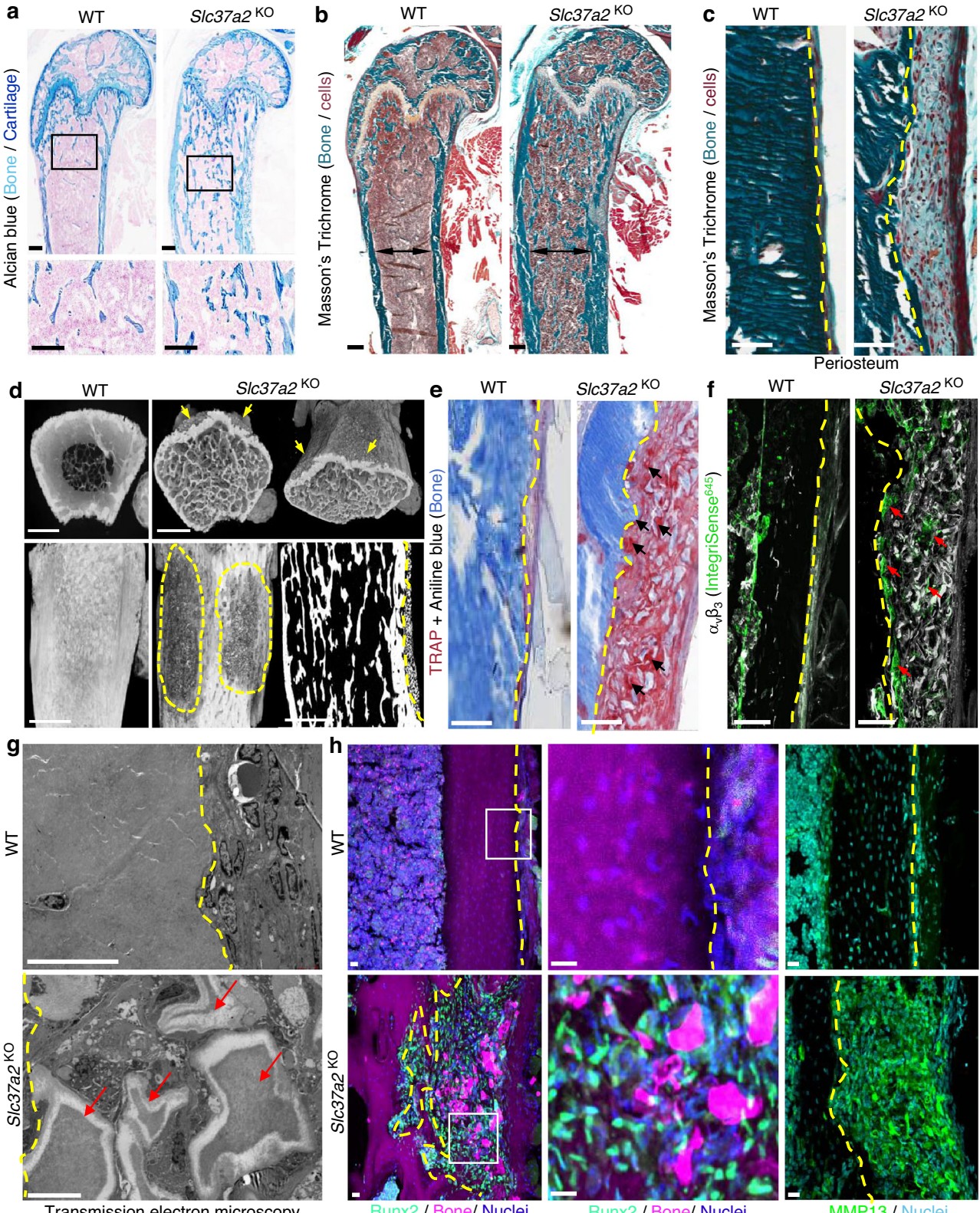

**Fig. 6 | *Slc37a2*[KO] mice exhibit high bone mass and a periosteal bone lesion.**
**a** Representative histological images of Alcian blue staining of the distal femur of 12-week-old female WT and *Slc37a2*[KO] mice (*n* = 5), Bars, 0.1 mm. **b** and **c** Mason's Trichrome staining of femurs from 12-week-old female WT and *Slc37a2*[KO] mice (**b**, *n* = 3) and representative images of the periosteal surface along the metaphyseal–diaphyseal junction (**c**, *n* = 3). Black double arrows indicate maximal femoral diameter and dashed lines indicate the periosteal bone surface. Bars, 1 mm. **d** High-resolution μCT images of 12-week-old female WT and *Slc37a2*[KO] femurs indicating the periosteal lesions (arrows and dashed outlines) (*n* = 2). Bars, 1 mm.

**e** and **f** Histological evaluation of TRAP[+] (**e**, *n* = 3) and αvβ3[+] (**f**, *n* = 3) cells along the periosteal surface of 12-week-old femurs from female WT and *Slc37a2*[KO] mice. Dashed lines denote the periosteal bone surface, arrows indicate TRAP[+] or αvβ3[+] cells. Bars, 1 mm. **g** Transmission electron microscopic evaluation of the periosteal layer of femurs from WT and *Slc37a2*[KO](*n* = 2). Dashed yellow lines indicate the periosteal bone surface. Red arrows depict demineralized bone/cartilaginous fragments. Bars, 10 μm. **h** Representative images of cryosections depicting the periosteal surface along the femurs of 12-week-old female WT and *Slc37a2*[KO] mice immunostained for Runx2 or MMP13 (*n* = 4). Bars, 20 μm.

these data indicate that Slc37a2 regulates the function, but not the differentiation, of osteoclasts in a cell-autonomous way.

### Slc37a2 deficiency impairs the export of monosaccharide sugars from SLs required for delivery to the ruffled border

To investigate the mechanism(s) by which Slc37a2 regulates osteoclast activity, comparative multi-omics analyses were performed on osteoclasts from WT and Slc37a2^KO mice (Supplementary Fig. 8). Except for the notable absence of Slc37a2, no consistent changes were detected both at the mRNA and protein levels between Slc37a2^KO and WT osteoclasts as determined by either RNAseq or label-free quantitative proteomics using a minimum of >1.5 $Log_2FC$ and an adjusted $P$ value of <0.05. Similarly, the proteomes of SLs isolated from Slc37a2^KO and WT osteoclasts were unremarkable (Supplementary Fig. 8). This suggests that Slc37a2 deficiency does not alter the global transcriptome or proteome of osteoclasts. On the other hand, quantitative metabolomic profiling of osteoclasts derived from WT and Slc37a2^KO mice revealed differences in the levels of several key metabolites, most notably monosaccharide sugars including D-(+)-glucose 1 and D-(−)-fructose1/2, which were increased in Slc37a2^KO osteoclasts compared to WT (Fig. 9a). The level of glucose-6-phosphate (G6P), the presumed substrate of Slc37a2[29], was however unaltered in Slc37a2-deficient osteoclasts.

It has long been known that glucose and fructose are transported out of lysosomes by an as-yet-unidentified monosaccharide transporter(s)[36]. Given that Slc37a2 resides on osteoclast SLs and that its absence coincided with increased intracellular levels of glucose and fructose, we considered the possibility that Slc37a2 regulates the export of monosaccharide sugars out of SLs. According to this scenario, SLs in Slc37a2^KO osteoclasts should be larger than those in controls due to osmotically imposed swelling upon substrate accumulation. Consistent with this premise, the size of SLs (co-labeled with LysoTracker green and DQ-BSA) in Slc37a2^KO osteoclasts were dramatically increased, and the associated network was less tubular when compared with their WT counterparts (Fig. 9b, c). Morphologically, Slc37a2^KO SLs appeared 'swollen' and resembled 'sucrosomes' (i.e. osmotically swollen endolysosomes that form upon the uptake and accumulation of undigestible sucrose)[37]. Addition of invertase, an enzyme that breaks down sucrose into its monosaccharide glucose and fructose constituents following its internalization and traffic into lysosomes, is known to induce the resolution (shrinkage) of sucrosomes, back to lysosomes, a process that coincides with extensive tubulation of lysosomal membranes[38,39]. Consistently, the addition of invertase (0.5 mg/ml, 1 h, S. cerevisiae) to osteoclast cultures superinfused with sucrose-containing solution (30 mM, 24 h) prompted rapid shrinkage and tubulation of WT LysoTracker Red⁺ sucrosomes (Fig. 9d) but failed to resolve sucrosomes in Slc37a2^KO osteoclasts when monitored over the same period by time-lapse confocal microscopy (Fig. 9e and Supplementary Movie 6). Thus, these data imply that Slc37a2 facilitates the export of glucose and fructose out of osteoclast SLs, a necessary prerequisite for the maintenance of SL size and the initiation of membrane tubulation.

In addition to initiating endomembrane remodeling events[40], the efflux of luminal solutes is an important requirement for lysosomal transport[41]. Therefore, we next asked whether the morphological disturbances observed in Slc37a2^KO SLs altered their capacity to translocate to the osteoclast-ruffled border. To this end, we employed immunofluorescent confocal microscopy to monitor the delivery of key SL marker proteins LAMP2 and Ctsk to the ruffled border in bone-resorbing osteoclasts. As shown in Fig. 9f–i, whereas WT osteoclasts exhibited prominent LAMP2 and Ctsk staining within ruffled borders demarcated by F-actin rings (white arrows), the signal intensity of LAMP2 and Ctsk was drastically reduced in Slc37a2^KO osteoclasts, indicative of impaired ruffled border delivery. As a consequence, the ruffled borders of Slc37a2^KO osteoclasts appeared underdeveloped and

immature when compared to the characteristic thin villous appearance of mature ruffled borders of WTs by TEM (Fig. 9j). Accordingly, ruffled border secretory activity was also diminished as indicated by the decreased levels of the cathepsin K cleavage product CTX-1 (Fig. 9k), as well as a reduction in the levels of the active form of MMP9 (Fig. 9l) when media was sampled from cultures of bone-resorbing Slc37a2^KO and WT osteoclasts. Altogether, these data suggest that, during bone resorption, Slc37a2 facilitates the export of monosaccharide sugars from SLs that, in turn, prompts SL tubulation and transport to the ventral plasma membrane for secretion, and thus maintenance of the osteoclast ruffled border.

## Discussion

Here we unmask the sugar transporter Slc37a2 as an obligatory component of osteoclast SLs and disclose several important findings that collectively advance our understanding of SL composition, organization, and regulation. First, we provide a proteomic account of the osteoclast SL, thus expanding the molecular inventory of this enigmatic organelle. Second, we establish Slc37a2 as a bona fide constituent of SL membranes, assigning a home to an otherwise orphan transporter. Third, we reveal that SLs exist as highly dynamic tubular organelles in living osteoclasts that fuse with the bone-lining plasma membrane and service the ruffled border, an observation that has previously gone unnoticed. Fourth, we provide detailed genetic evidence that Slc37a2 is essential for osteoclast function and bone metabolism, thus promoting Slc37a2 to the league of osteoclast transporters linked to the pathophysiological regulation of bone[42]. Finally, our work establishes Slc37a2 as a candidate SL monosaccharide exporter, addressing a long-standing question in the lysosome field.

Originally identified as a cAMP-inducible gene in RAW264.7 macrophages[43], Slc37a2 is considered a member of the Slc37 subfamily (Slc37a1–a4) of transporters which reside in the endoplasmic reticulum (ER) and function as Pi-linked antiporters for sugar-phosphates such as G6P[29]. However, at least two lines of mechanistic and structural evidence hint that Slc37a2 is unlikely an ER-anchored G6P transporter. First, unlike the archetype Slc37 family member Slc37a4, which functions as a Pi-linked G6P antiporter coupled to G6Pases-a/b, the exchange activity of Slc37a2 is insensitive to the G6Pases inhibitor chlorogenic acid, indicating that Slc37a2 is not a physiological G6P transporter[44]. Second, whereas Slc37a4 exists as a 10-transmembrane spanning protein localized to the endoplasmic reticulum (ER)[45], Slc37a2 possesses 12 transmembrane segments housing an extended N-terminal loop that undergoes extensive N-linked glycosylation[46] as well as localization signals (e.g. YxxØ) within its extreme C-terminus, criteria usually reserved for endolysosomal membrane proteins. Indeed, Slc37a2 has previously been detected on lysosomal membranes isolated from the liver by proteomics[47]. In addition, during the course of this work, Slc37a2 was reported to localize to an endolysosomal-related organelle dubbed the 'gastrosome' in zebrafish microglia[48]. Taken together with our extensive biochemical, morphological, and functional studies, which localize Slc37a2 to SLs in osteoclasts, we posit that the Slc37a2 transporter is a membrane resident of endolysosome-related organelles.

A remarkable feature of Slc37a2-bearing SLs is their extremely dynamic nature and ability to extend and maintain elongated tubules that fuse with the osteoclast bone-lining plasma membrane. What is the role of 'tubular' SLs in osteoclasts during bone resorption? Based on our live cell results, we favor a model whereby SLs undergo microtubule-coupled extension and tubulation necessary to infiltrate the narrow infolds of the ruffled border for cargo delivery (Fig. 10). Although our 'tubular SL' model challenges the traditional paradigm of ruffled border genesis i.e. via the fusion of acidified 'vesicles'[16], it offers a simple yet plausible explanation to reconcile how SLs (typically

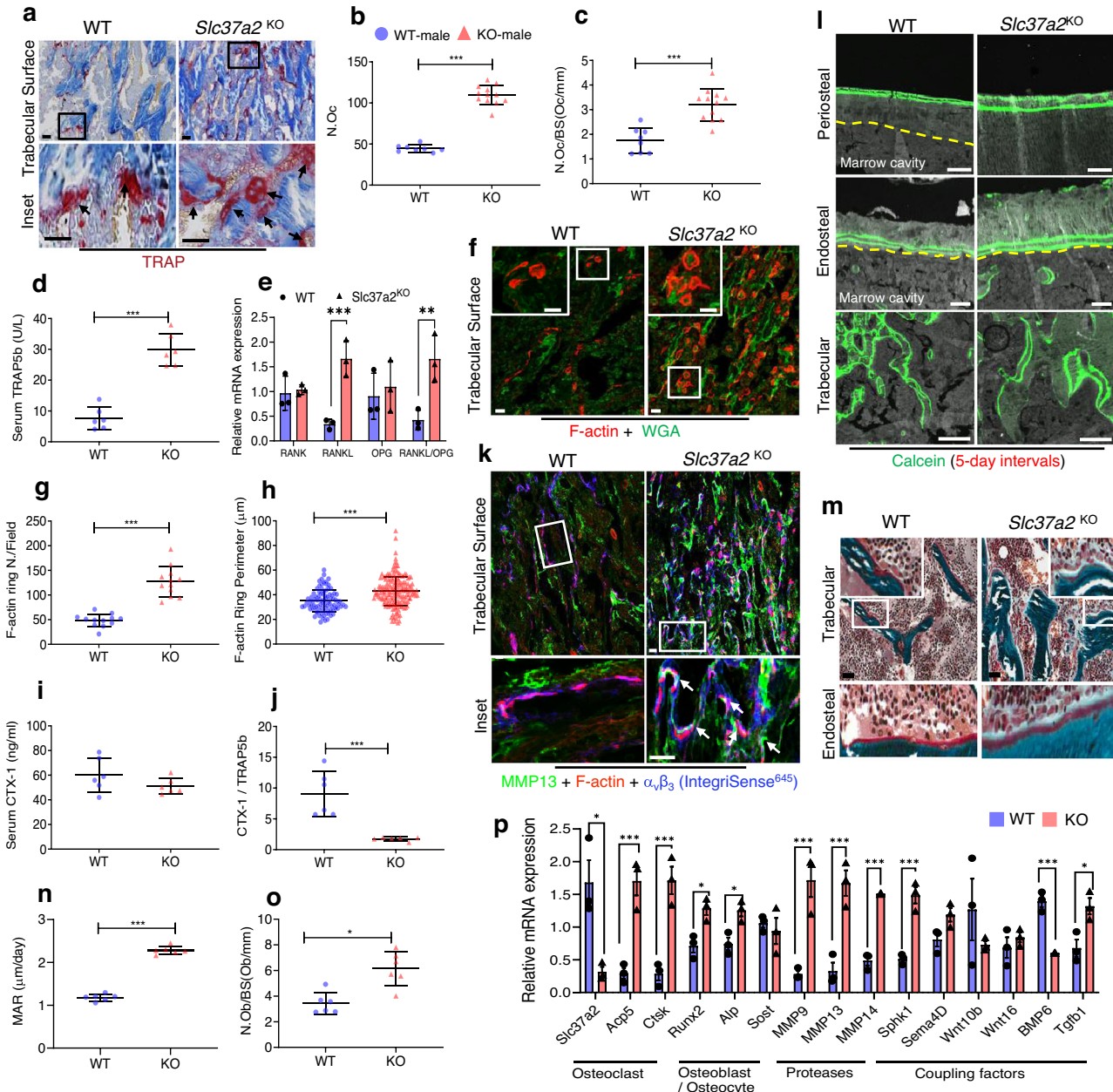

**Fig. 7 | *Slc37a2* deletion impairs bone resorption leading to compensatory increases in osteoclastogenesis, protease expression, and osteoblast-mediated bone formation. a** Representative images of the primary spongiosa of proximal WT and *Slc37a2*KO male femora stained for TRAP and **b, c** histomorphometric analysis of TRAP+ osteoclasts (*n* = 8 WT and 12 *Slc37a2*KO animals per group) (**b** *P* < 0.001; **c** *P* < 0.001). Bar, 50 μm. **d** Serum TRAP5b levels (*n* = 6 animals per genotype, *P* < 0.001) and **e** RANK/RANKL/OPG mRNA expression in femurs (*n* = 3 animals per genotype, *RANK P* = 0.750121; *RANKL P* = 0.003713; *OPG P* = 0.658277; *RANKL/OPG P* = 0.014159*)* of 12-week-old male WT and *Slc37a2*KO mice. **f–h** Confocal images of representative femur cryosections stained for F-actin and Wheat-germ agglutinin, (WGA) from 12-week-old male mice (**f**) and quantitation of F-actin numbers (**g**, *n* = 12 fields/group *P* < 0.001) and diameters (**h**, *n* = 92 cells/5 independent WT samples and 192 cells/9 independent *Slc37a2*KOsamples, *P* < 0.001). Bars 50 μm. **i** CTX-1 levels (*n* = 6, *P* = 0.1815) and **j** ratio of CTX-1/TRAP5b levels in serum of 12-week-old male WT and *Slc37a2*KO mice (*n* = 6, *P* = 0.0006).

**k** Representative confocal image of the primary spongiosa of WT and *Slc37a2*KO male femora immunostained for MMP13, F-actin and αvβ3 (IntegriSense645). Arrows indicate flat bone-lining MMP13+ cells occupying resorptive lacunae in magnified images. Bars 50 μm. **l–o** Representative images of 12-week-old male femurs from WT and *Slc37a2*KO mice depicting fluorescent calcein-double labeling (**l**, Bars, 100 μm), Mason's Trichrome staining (**m**, Bars, 100 μm) and histomorphometric analyses of bone mineral apposition rate (**n**, *n* = 6, *P* < 0.001) and osteoblast numbers/ bone surface (**o**, *n* = 6, *P* = 0.0186). **p** Relative mRNA expression in femora from 12-week-old WT and *Slc37a2*KO mice (*n* = 3). *Slc37a2 P* = 0.017655; *Acp5 P* = 0.003896; *Ctsk P* = 0.003602; *Runx2 P* = 0.016348; *Alp P* = 0.018260; *MMP9 P* = 0.005514; *MMP13 P* = 0.004579; *MMP14 P* = 0.000163; *Sphk1 P* = 0.001930; *BMP6 P* = 0.000829; *Tgfb1 P* = 0.023799. All data are presented as means ± SD. *P < 0.05, ** P < 0.01, ***P < 0.001 by two-tailed unpaired Student's *t* tests. Source data are available in the Source Data file.

200–900 nm in diameter)[49] are able to adapt to overcome the size constraints imposed by the tight membrane folds of the ruffled border (~70–400 nm in diameter)[50]. This would also provide a mechanism for how collagenolytic enzymes, such as cathepsin K and MMP9, are

continuously delivered to and secreted by the ruffled border during extended periods of bone resorption, such as occur during the formation of extended 'trench-like' resorptive trails[51]. Indeed, MMP9 secretion and activity have recently been shown to be mediated

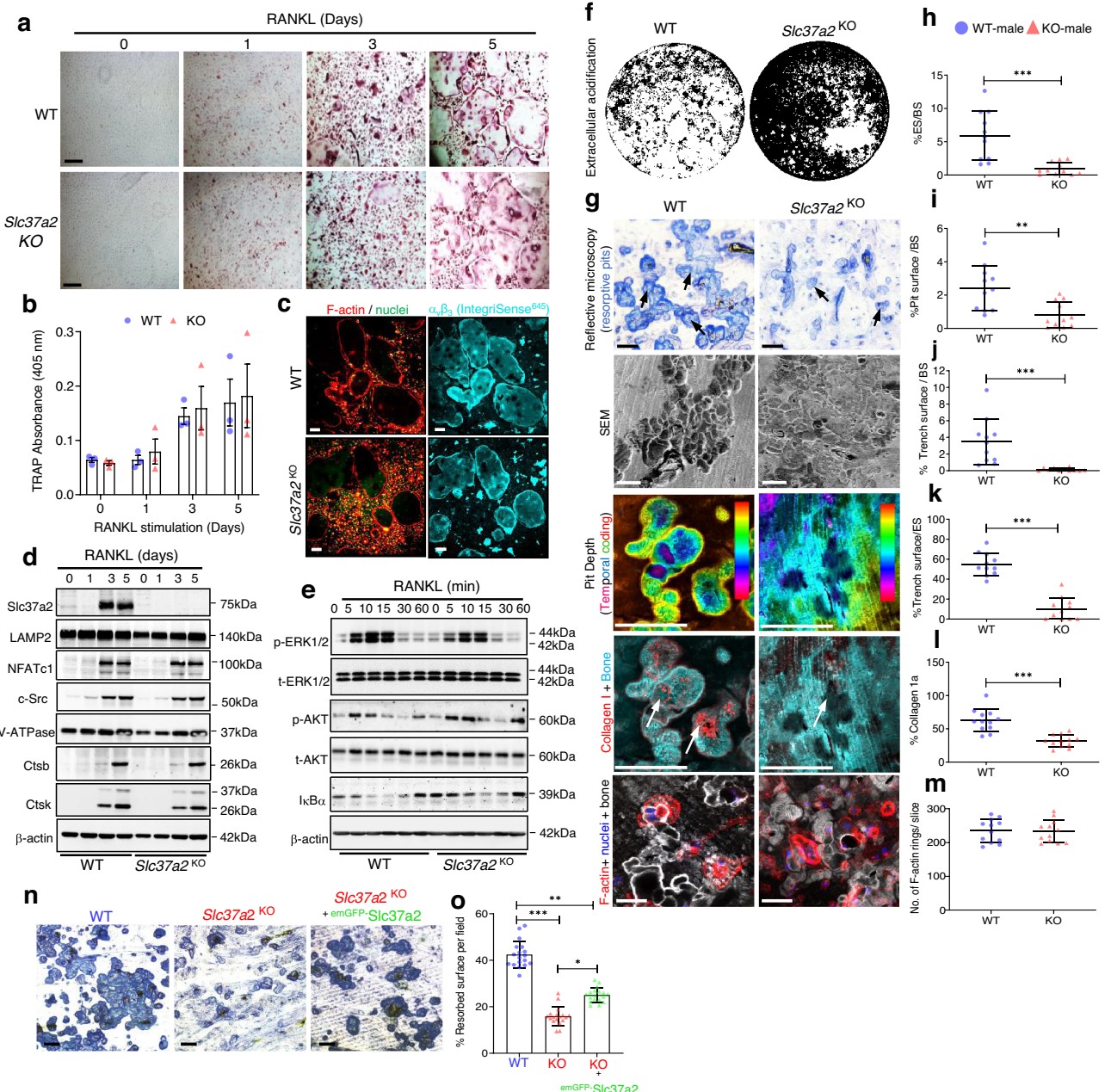

**Fig. 8 | *Slc37a2* deletion impairs bone resorption but does not alter osteoclast differentiation, spreading, or RANKL-signaling cascades. a** and **b** WT and *Slc37a2^KO* BMM-derived osteoclasts were TRAP stained (**a**), and TRAP activity measured (405 nm) (**b**) (*n* = 3). Bar, 50 μm. **c** Representative images of WT and *Slc37a2^KO* osteoclasts immunostained for F-actin, αvβ3 (IntegriSense^645) and nuclei. Bar 50 μm. **d** Expression of osteoclast and SL markers during RANKL-induced differentiation of WT and *Slc37a2^KO* BMMs by immunoblotting using indicated antibodies (*n* = 3). **e** RANKL-signaling in WT and *Slc37a2^KO* BMMs. Immunoblotting of the levels and phosphorylation states of ERK1/2, AKT, and IκBα in response to RANKL (*n* = 3). **f** Representative Von Kossa staining of WT and *Slc37a2^KO* osteoclasts on mineralized substrates (*n* = 3). **g**–**m** Representative images of in vitro bone resorption assays of WT and *Slc37a2^KO* osteoclasts using reflective microscopy, SEM, and confocal microscopy with quantification of the percentage of eroded surface (**h**, *n* = 11 bone discs from 3 experiments; *P* = 0.0004), pit area (**i**, *n* = 11 bone discs from 3 experiments; *P* = 0.0029), trench surface per bone slice (**j**, *n* = 11 bone

discs from 3 experiments; *P* = 0.0006), trench surface/eroded surface (**k**, *n* = 11 bone discs from 3 experiments; *P* < 0.0001), collagen1a staining (**l**, *n* = 12 bone discs from 3 experiments; *P* < 0.0001) and a number of F-actin rings per bone slice (**m**, *n* = 12 bone discs from **m** 3 experiments; *P* = 0.9293). Black and white arrows indicate bone resorption pits in toluidine blue and collagen 1 stained bone slices, respectively. Bars, 50 μm. **n** and **o** Rescue of *Slc37a2^KO* osteoclasts activity by lentiviral transduction of ^emGFP-Slc37a2. Depicted are representative images of toluidine blue stained resorptive pits of WT, *Slc37a2^KO*, and ^emGFP-Slc37a2 transduced *Slc37a2^KO* osteoclasts with quantification of the percentage resorbed surface per field quantified (*n* = 54 fields from 3 experiments, WT vs. Slc37a2^KO *P* < 0.0001; WT vs. ^emGFP-Slc37a2^KO *P* = 0.0011; Slc37a2^KO vs. ^emGFP-Slc37a2^KO *P* = 0.0250). All data are presented as means ± SD. * *P* < 0.05, ** *P* < 0.01, *** *P* < 0.001 by two-tailed unpaired Student's *t*-test (**h**–**m**) or nested one-way ANOVA (**o**). Bar, 50 μm. Source data are available in the Source Data file.

via tubular lysosomes in macrophages[52]. Alternatively, tubular SLs may serve to deliver components of the bone resorptive machinery to the ventral plasmalemma. The 116 kDa *a3* isoform (Tcirg1) of the V-ATPase proton pump, for example, is known to translocate from lysosomes to

the bone-oriented osteoclast plasma membrane[12] and has been shown, in macrophages, to be transported via tubular lysosomes to phagosomes to establish the acidic environment hostile to pathogens[53]. Given the analogies shared between the ruffled border

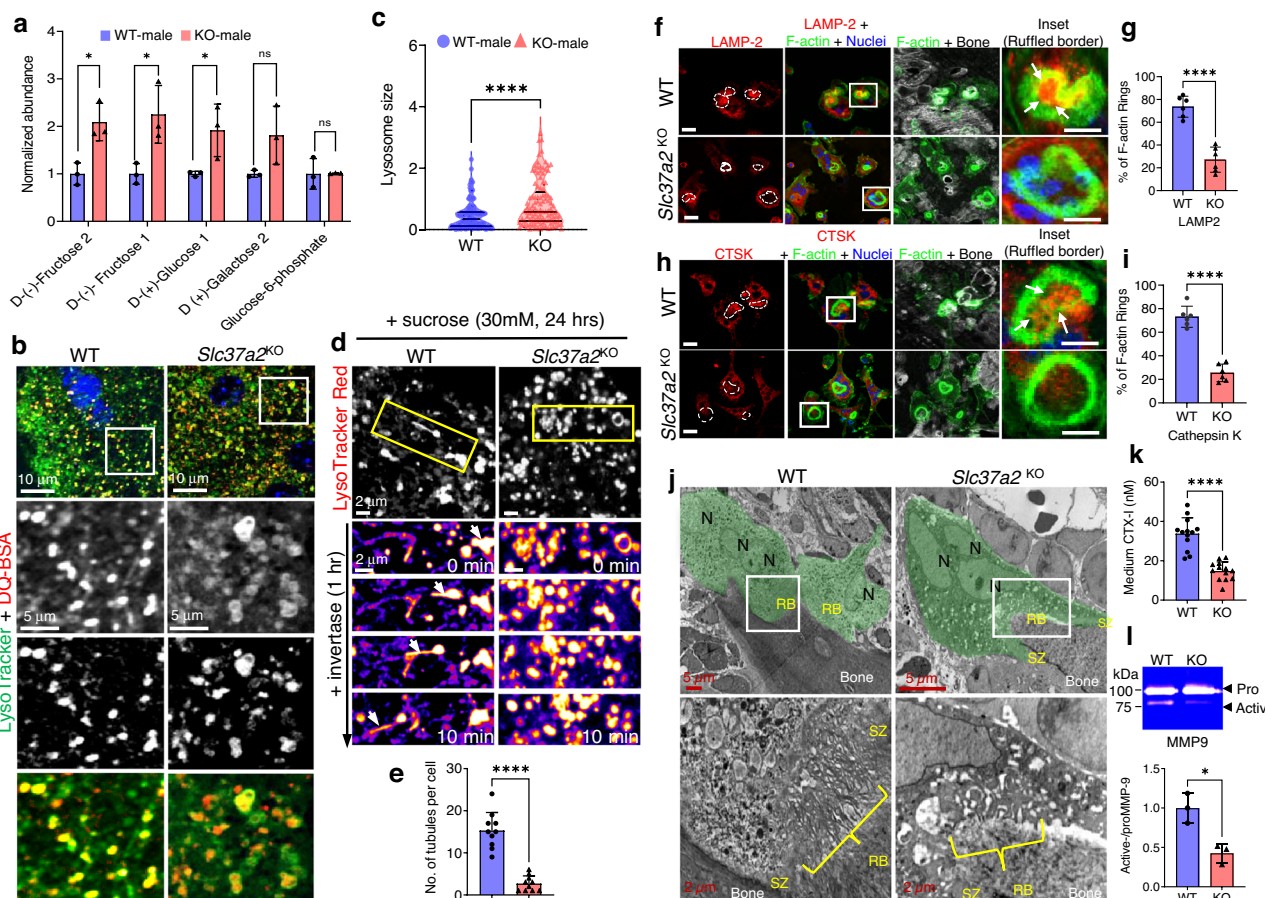

**Fig. 9 | Export of monosaccharides and delivery of SLs to the ruffled border is impaired Slc37a2$^{KO}$ osteoclasts. a** Quantitative metabolomics profiling of monosaccharide sugars in BMM-derived osteoclasts from WT and Slc37a2$^{KO}$ mice ($n = 3$; D-(−)-Fructose 2 $P = 0.014536$; D-(−)-Fructose 1 $P = 0.028321$; D-(+)-Glucose 1 $P = 0.046169$). **b** and **c** Representative confocal images of WT and Slc37a2$^{KO}$ osteoclasts stained with LysoTracker Green and DQ-BSA (**b**) and quantitation of SL size (**c**) ($n = 109$ SLs pooled from 10 cells per group, $P < 0.0001$). **d** Time-lapse confocal image series of WT and Slc37a2$^{KO}$ osteoclasts cultured in high sucrose conditions (30 mM, 24 h) to induce sucrosome formation and imaged for 10 min post-treatment with invertase (0.5 mg/ml, 1 h). SLs were pulsed with LysoTracker Red prior to imaging. Arrows track an SL resolution and tubulation event. **e** Number of tubules per cell post-treatment with invertase ($n = 10$ cells, $P < 0.0001$). **f**–**i**

Confocal images of WT and Slc37a2$^{KO}$ osteoclasts cultured on bone and immunostained for F-actin in combination with either LAMP-2 (**f**, Bars, 10 μm) or cathepsin k (CTSK) (**h**, Bars, 10 μm) and quantitation (**g**, $P < 0.0001$; **i**, $P < 0.0001$). Arrows indicate delivery of LAMP2 and CTSK within F-actin rings ($n = 6$). **j** Representative TEM micrographs of WT and Slc37a2$^{KO}$ osteoclasts (green) lining trabecular bone surfaces within the primary spongiosa of 5-day-old male littermates ($n = 3$). Magnified pictures illustrate ruffled borders (RB). N nuclei, SZ sealing zone. **k, l** CTX-1 and active MMP9 levels in media from cultures of bone-resorbing WT and Slc37a2$^{KO}$ osteoclasts as monitored by ELISA (**k**) and gel zymography (**l**). $n = 13$, $P < 0.0001$ (**k**) and $n = 3$, $P = 0.0117$ (**l**). Data are presented as means ± SD *$P < 0.05$ and ****$P < 0.0001$ by two-tailed unpaired Student's t-test. Source data are available in the Source Data file. See also related Supplementary Movie 6.

and phagolysosomal membranes, it is therefore conceivable that a3 and/or other endolysosomal proteins (e.g. LAMP2) may be delivered to the osteoclast ventral plasma membrane via tubular SLs. Such a model would be most consistent with the: (i) impaired delivery of LAMP2 and cathepsin K to the ruffled border membrane; (ii) reduced extracellular secretion of cathepsin K and active-MMP9; (iii) altered ruffled border morphology and; thus (iv) resorptive deficits exhibited by Slc37a2$^{KO}$ osteoclasts.

Perhaps the most striking observation of this study is the high bone mass phenotype in Slc37a2$^{KO}$ mice. This phenotype is attributed primarily to impaired bone resorption by osteoclasts leading to imbalanced remodeling-based bone formation by osteoblasts. In vivo, this deficit in osteoclast function is paralleled by enhanced osteoclastogenesis via increased RANKL/OPG levels, as well as increased expression of proteases known to participate in collagen-matrix digestion (e.g. MMP9, MMP13, MMP14)[54,55] and key coupling factors that promote remodeling-based bone deposition by osteoblasts (e.g. Sphk1/S1P and TGF-β)[35]. Altogether, these phenotypic and cellular features are strikingly reminiscent of those reported for

cathepsin K-deficient mice[56–58]. What distinguishes Slc37a2$^{KO}$ mice from mice lacking cathepsin K, and other models of osteosclerosis attributed to osteoclast dysfunction e.g. β3-integrin[59] and Plekhm1[60], is the conspicuous fibro-cartilaginous lesion observed within the physeal–metaphyseal interface. While the pathoetiology of this lesion requires further study, it may reflect the failure of Slc37a2$^{KO}$ osteoclasts to complete bone 'cut-back' i.e. to decrease bone width within the periosteal metaphyseal area during longitudinal growth[61]. Alternatively, it may point to disturbances in a population of matrix-degrading cells specific to the periosteal-bone niche that are distinct from osteoclasts but are functionally dependent on Slc37a2. Future generation and characterization of cell-specific conditional Slc37a2 knockout models may help to discern this.

Previous biochemical studies have described the transport of glucose and fructose out of lysosomes by unknown monosaccharide transporters[36]. Our results suggest that, in osteoclasts, Slc37a2 fulfils this role. First, metabolic profiling demonstrated that loss of Slc37a2 led to increased levels of glucose and fructose in osteoclasts. Second,

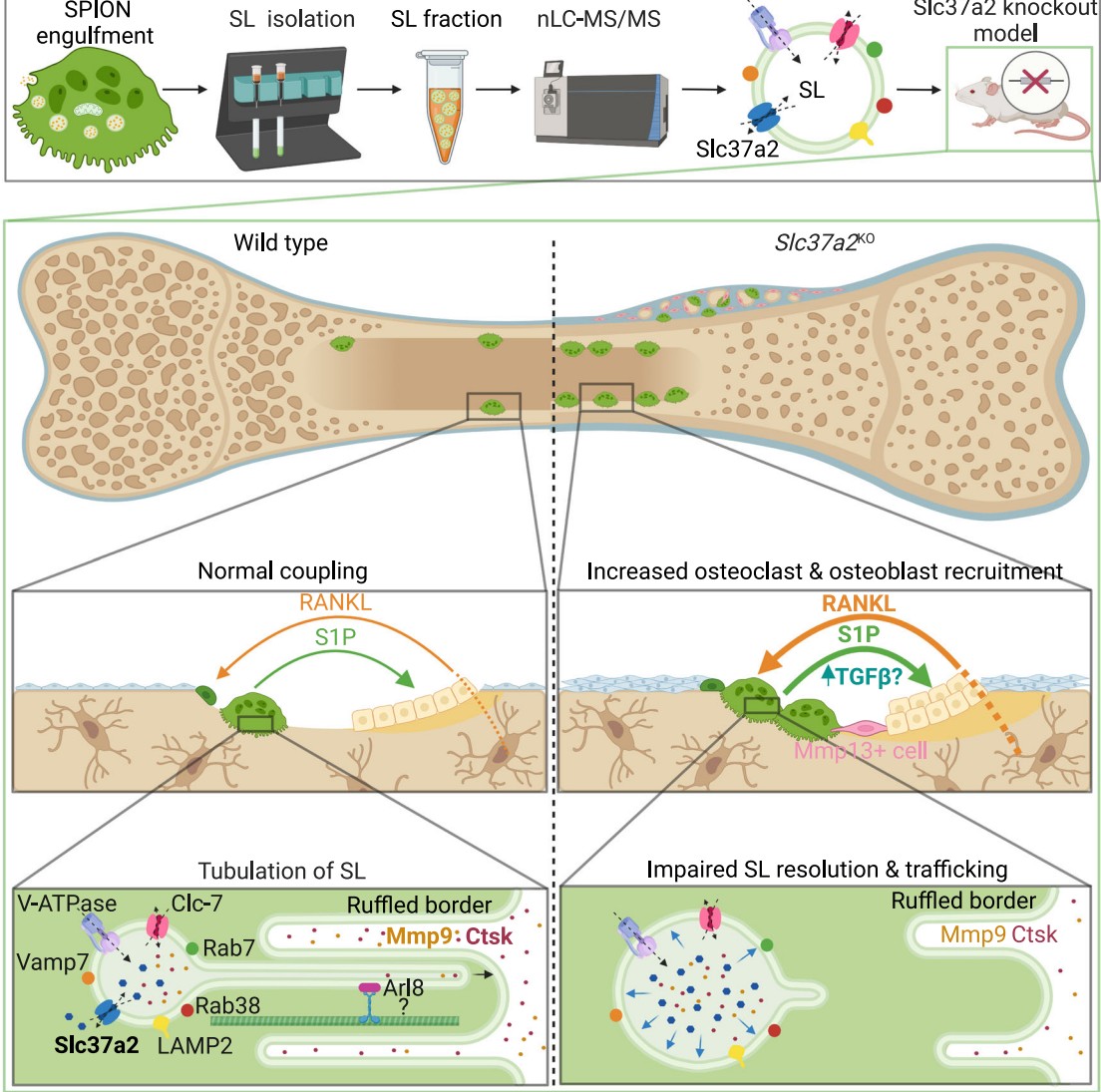

**Fig. 10 | Working model of Slc37a2 in osteoclast function and bone metabolism.** Top panel provides an overview of proteomic identification of sugar transporter Slc37a2 on osteoclast secretory lysosomes, the middle panels highlight tissue level alterations in osteoclast–osteoblast coupling during bone remodeling in *Slc37a2*-deficient mice and the bottom panel illustrates the proposed role for Slc37a2 in the regulation of sugar export from secretory lysosomes, a prerequisite for secretory lysosome resolution, membrane tubulation and delivery to the osteoclast ruffled border. Created with BioRender.com.

the morphological swelling of SLs in *Slc37a2*KO osteoclasts together with the impaired capacity of 'sucrosomes' to undergo invertase-induced resolution, indicates that the export of glucose and fructose from SLs is inhibited when Slc37a2 is lacking, in keeping with a lysosomal storage disorder. Considering that the extrusion of monosaccharides is an established prerequisite for lysosomal resolution[38], tubulation[39], and transport[41], we envisage a membrane remodeling cascade whereby Slc37a2-mediated export of glucose and fructose drives the osmotic extrusion of water from SLs, which in turn, reduces their hydrostatic tension, thus prompting SL tubulation and transport to the bone-lining plasmalemma and/or ruffled border (Fig. 10).

In recent years, the critical importance of SLCs in human metabolic diseases and their largely untapped potential as 'druggable' targets has gained increasing momentum[62–64]. Altogether, our findings that *Slc37a2* deletion increases skeletal bone mass with age and attenuates menopause (OVX)-associated trabecular bone loss, lends a rationale for the development of Slc37a2 inhibitors as a potential therapeutic treatment for metabolic bone diseases.

## Methods

This research complies with the Animal Welfare Act 2002 (WA) and requirements of the eighth (2013) edition of the Australian code for the care and use of animals for scientific purposes. All protocols involving animals were approved by the Animal Ethics Committee of The University of Western Australia (Approval No. RA13/100/1475).

### Animals

Slc37a2 gene-trapped ES cells (JM8A3.N1 cell line, Clone ID EPD0641_4_E06) were obtained from the KOMP consortium and provided by the UC Davis KOMP Repository. Slc37a2 knockout (KO) mice were generated through the Australian Phenomics ES Cell-to-Mouse Service (Monash University). All mice were maintained on the C57BL/6N background under specific-pathogen-free conditions and routinely monitored by animal care staff at the Animal Resources Centre (ARC), Murdoch, Western Australia (WA). Mice were housed in standard cages (45 × 29 × 12 cm) and maintained at a temperature of 22 ± 1 °C and a humidity-controlled room (40–65%) on a 12 h light cycle with ad libitum access to water

and a standard laboratory chow diet (SF00-100, Specialty Feeds, Glen Forrest, WA). Mice were euthanized by cervical dislocation under inhalation anesthesia (Fluriso, Isoflurane USP, VetOne).

### Genotyping

The genotype of mice was determined by PCR using the cycling parameters recommended by UC Davis KOMP Repository. The presence of Slc37a2 WT and tm2a alleles in the mice genome were analyzed by multiplex PCR using primer sequences CSD-loxF: GAGATGGCGCAACGCAATTAATG, CSD-R: GACCTAGCTTCTCAGGCATCTCAGG, CSD-Forward: AGAAGGGGTACAGAAGCAACAGC, CSD-ttReverse: CCATGTGTCCCTTCATGCTTTAGG.

### Bone marrow macrophage extraction and osteoclast culture

Bone marrow macrophages (BMMs) were harvested from the long bones of Slc37a2$^{+/+}$ wild-type (WT) and Slc37a2$^{-/-}$ (KO) mice in accordance with the UWA Institutional Animal Ethics Committee Guidelines. Unless otherwise specified, all cell culture reagents were purchased from Thermo Fisher Scientific. Briefly, extracted primary cells were cultured in complete α-MEM containing 10% fetal bovine serum (FBS), 2 mM L-glutamine, 50 U/ml penicillin, 50 U/ml streptomycin, and supplemented with M-CSF (25 ng/ml). All cultures were maintained in humidified conditions of 5% $CO_2$ at 37 °C. Osteoclasts were generated from BMMs differentiated with RANKL (10 ng/ml, R&D Systems) and M-CSF (2 ng/ml) in vitro. Cells were then harvested and processed for RNA extraction, immunoblot or proteomic analysis, or seeded onto glass coverslips, glass-bottomed culture dishes (ibidi), cortical bone slices (Boneslices.com), or Osteo Assay Surface Stripwells (Corning).

### Secretory lysosome extraction

Secretory lysosome (SL) extraction was adapted using methods described previously[17,18], but with the following modifications. Briefly, large-scale osteoclastogenesis was performed on $4 \times 15$ cm diameter dishes at a cell density of $5 \times 10^6$ BMMs per plate. Following osteoclast formation, the cells were 'pulsed' for 24 h with 10 μl of superparamagnetic iron oxide nanoparticles (SPIONs) (carboxyl-Adembeads; Ademtech) per dish and 'chased' for an additional 24 h to accumulate SPIONs in secretory lysosomes. Cells were washed with warm PBS and scraped in homogenization buffer (10 mM HEPES–KOH pH 7.4, 250 mM sucrose, 5 mM DTT, 1 mM EDTA, and protease inhibitor cocktail, Roche). The cells were then homogenized using an Eberbach Corporation (E2355) Con-Torque Tissue Homogenizer at the highest rpm setting, a PYREX® 5 ml Potter-Elvehjem and Pestle (7725T-5) for 30 strokes then passed 8 times through a 23 G needle. Lysates were collected and centrifuged at $200 \times g$ for 10 min at 4 °C to remove debris and unbroken cells. The supernatants were then loaded onto a Miltenyi Biotec MS Column (130-042-201) equilibrated with 1 ml 0.5% BSA in PBS and immobilized SPIONs washed with DNase solution (1 mg/ml DNase, 0.1 mM sucrose in PBS). Fractions (F1–F3) containing enriched SLs were collected by separating the column from the Miltenyi Biotec MiniMACS™ Separator magnet and sequentially eluting the column with 200 μl of elution buffer (0.1 mM sucrose, protease inhibitor cocktail in PBS). Aliquots of the three enriched SL fractions (F1–F3), the cell homogenate (H), the post-nuclear supernatant (PNS), and the column flow-through (UB) were snap frozen in liquid $N_2$ and stored at −80 °C until required. Successful isolation of SLs was confirmed using immunoblotting for specific organelle markers, transmission electron microscopy, and an enzymatic assay for β-Hexosaminidase activity.

### Proteomic sample preparation

Protein concentration of fractions was determined by Bradford assay (Bio-Rad #5000006). 10 μg of the total cell homogenates and SL fraction F2 (each corresponding to a biological triplicate) were used for proteomic analysis. Briefly, samples were mixed with SDS–PAGE sample buffer and denatured before 1-dimensional electrophoresis (1-DE) through an Invitrogen precast 10% NuPAGE Bis-Tris gel with corresponding NuPAGE running buffer at 80 V for 1 h. The gel was washed with Milli-Q $H_2O$ and fixed with 40% ethanol, and 10% acetic acid in Milli-Q $H_2O$ for 15 min with gentle rocking. After a series of Milli-Q $H_2O$ washes, the gel was stained using a pre-prepared Bio-Rad Colloidal Coomassie stain (#1610803) and then de-stained with Milli-Q $H_2O$. The gel lanes were then divided into 10 fractions. The 10 gel fractions were further divided into 1 mm square cubes, further de-stained, and then digested in-gel with trypsin to extract peptides according to the methods of Shevchenko et al.[65]. Peptides were dried using a Speed-Vac and analyzed by mass spectrometry.

### LC–MS/MS analysis

For mass spectrometry analysis, peptides were separated at a flow rate of 300 nl/min on a 50 cm long, 75 μm internal diameter EASY-spray PepMap C18 column (Thermo Fisher Scientific) using a Dionex UltiMate 3000 Nano-UHPLC system (Thermo Fisher Scientific). The column was maintained at 50 °C. Buffers A and B were 0.1% formic acid in water and 0.1% formic acid in acetonitrile, respectively. Peptides were eluted in 165 min runs, with a gradient from 3% to 5% buffer B for 5 min, from 5% to 25% buffer B for 105 min, from 25% to 35% buffer B for 15 min, from 35% to 95% buffer B for 20 min, and equilibration (3% buffer B) for 19 min. Eluting peptides were analyzed on an Orbitrap Fusion mass spectrometer (Thermo Fisher Scientific). The instrument was operated in a data-dependent, "Top Speed" mode, with cycle times of 3 s. Peptide precursor mass to charge ratio ($m/z$) measurements (MS1) were carried out at 120,000 resolution in the 375–1500 $m/z$ range. The MS1 AGC target was set to 4E$^5$ and the maximum injection time to 50 ms. Precursor priority was set to "most intense" and precursors with charge states 2–7 only were selected for HCD fragmentation. Fragmentation was carried out using 35% collision energy. The $m/z$ of the peptide fragments were measured in the ion trap using an AGC target of 1E$^4$ and 50 ms maximum injection time.

### Data analysis

Mass spectrometry spectral data were analyzed using Proteome Discoverer™ 2.3 Software (Thermo Fisher Scientific). Briefly, the reference FASTA library used for analysis was downloaded from UniProtKB (5/02/2020) for *Mus musculus* containing 86,447 reviewed and unreviewed peptides. The cleavage sites were set to correspond to Trypsin and the maximum missed cleavage sites were set at 2. Minimum and maximum peptide lengths were set at 7 and 144, respectively, with the precursor mass tolerance set at 4.4 ppm. The biological triplicate peptide abundances were pooled and abundance ratios were calculated for total cell homogenate pooled triplicates over SL (F2) pooled triplicates.

### Analysis of human orthologues

Enrichment of genome-wide association study (GWAS) associations among human homologs of the mouse SL gene set was performed using the VEGAS2Pathway function of the VErsatile Gene-based Association Study 2 (VEGAS2) software package[24]. The VEGAS2Pathway approach tests for the enrichment of significant associations in GWAS summary data in pre-defined gene sets, while correcting for gene lengths, linkage disequilibrium (LD) between markers, and the size of the gene set. Gene analysis windows included 50 kb on either side of each gene, with the 1000 Genomes phase 3 EUR dataset used as the LD reference. A custom gene set was created for the analysis consisting of the human homologs of the top 218 enriched mouse genes identified in our studies. The estimated bone mineral density (eBMD) $P$ values were generated using a linear mixed non-infinitesimal model in the BOLT-LMM v2 software package.

## Generation of Slc37a2 antibodies

A Slc37a2-specific polyclonal antibody was generated by immunizing rabbits with the peptide 'CTPPRHHDDPEKEQ,' corresponding to the cytoplasmic facing intracellular loop of mouse Slc37a2. Antisera were affinity-purified against the corresponding immunization peptide. Slc37a2 peptide antibodies were produced using the PolyExpress™ service by GenScript (New Jersey, USA). Antibodies were validated against Slc37a2^KO osteoclasts by immunostaining (1:100 dilution) and immunoblotting (1:1000 dilution) (see Supplementary Fig. 1).

## Additional antibodies

For details of additional commercial antibodies used in the "Methods" section see Supplementary Table 1.

## Sample preparation for cryosectioning and immunohistochemistry

Hindlimbs of mice were dissected and muscles were thoroughly removed before the femurs were fixed in 10% formalin at 4 °C overnight. Samples were washed thrice with PBS before decalcification in cold, 14% EDTA pH 8.0 for 48 h. Decalcified bone samples were incubated in 15% and 30% sucrose (Sigma-Aldrich) in PBS each for 48 h at 4 °C, then embedded in SCEM-L1 embedding medium (SECTION-LAB), and rapidly cooled using a hexane-dry ice coolant mix. Bone cryosections (20–30 μm) were generated with a tungsten carbide blade (ProSciTech) and Leica CM1900 cryostat, using Kawamoto's film method[66].

For immunostaining, cryosections were completely thawed at RT and rehydrated with PBS at RT for 5 min to remove the embedding medium. Sections were permeabilized with 0.3% Triton-X/PBS at RT for 20 min and blocked with 5% BSA/PBS at RT for 30 min. Primary antibodies were incubated at 4 °C overnight. Primary antibodies included: Slc37a2 (rabbit polyclonal, this paper, 1:100 dilution), RUNX2 (rabbit monoclonal, D1L7F, Cell Signaling, 1:200 dilution), MMP13 (rabbit polyclonal, Abcam, ab39012 1:300 dilution). Conjugate antibodies and fluorescent probes included: Rhodamine phalloidin (Thermo Fisher Scientific, R415, 1:500 dilution), Alexa Fluor 647 Phalloidin (Thermo Fisher Scientific, A22287, 1:300 dilution), IVISense Osteo 680 (PerkinElmer, NEV10020EX, 1:300 dilution), IVISense Integrin Receptor (IntegriSense™) 645 (PerkinElmer, NEV10640, 1:300 dilution). Following primary antibody incubation, sections were washed and incubated with secondary antibodies at RT for 2 h. Secondary antibodies used included: anti-rabbit Alexa Fluor 488 (Thermo Fisher, A-11029, 1:500 dilution), and nuclei were stained using Hoechst 33258 (Cat. no. H3596, Invitrogen, 2 μg/ml). Sections were washed, mounted onto glass slides using ProLong Glass antifade mountant (Cat. no. P36980, Invitrogen) and sealed with glass coverslips, and imaged by confocal microscopy. Sections were imaged using the NIKON A1R in a Ti-E inverted motorized confocal microscope equipped with Nikon PlanApo ×20 (air), Nikon PlanApo ×60 WI IR, 1.27NA (oil), and Nikon PlanApo ×100, 1.45 NA (oil) immersion lenses.

## Protein extraction and immunoblotting

Cells were lysed on ice in RIPA buffer supplemented with Protease Inhibitor Cocktail (Cat#. 11873580001, Roche). Lysates were cleared by centrifuging at 15,000×*g* and 4 °C for 20 min (Sorvall Legend Micro 17R Centrifuge, Cat#. 75002440, Thermo Fisher Scientific). Protein concentrations were determined through Bradford assays (Cat#. 5000006, Bio-Rad). For immunoblotting, a loading buffer was added and samples were incubated either at 37 °C for 1 h or 99 °C for 5 min before SDS–PAGE electrophoresis and transfer onto nitrocellulose membranes. Membranes were blocked with 5% skim milk in TBST, and primary antibodies were incubated overnight at 4 °C in TBST with 1% skim milk. Primary antibodies include: LAMP-2 (DSHB ABL-93, 1:1000 dilution), Rab7 (E9O7E, Cell Signaling 95746, 1:2000 dilution), Ctsk (Santa Cruz sc-48353, 1:250 dilution), Rab38 (Santa Cruz sc-390176,

1:200 dilution), Rab5 (Synaptic Systems 108 011, 1:1000 dilution), Syntaxin 16 (Stx16) (Synaptic Systems 110 163, 1:300 dilution), Rab1b (Santa Cruz sc-599, 1:5000 dilution), VDAC3 (Santa Cruz sc-292328, 1:500 dilution), VGluT1 (Synaptic Systems 135 303, 1:1000 dilution), Slc37a2 (this paper, 1:1000 dilution), V-ATPase D1 (Santa Cruz sc-81887, 1:1000 dilution), NFATc1 (DSHB, 7A6 1:1000 dilution), c-Src (Millipore Merck: 05-184, clone GD11, 1:1000 dilution), Ctsb (Atlas Antibodies, Sigma Aldrich HPA018156, 1:250 dilution), pERK1/2 (Santa Cruz sc-7383, 1:500 dilution), t-ERK1/2 (Promega V114A, 1:1000 dilution), p-AKT (Ser473) (Cell Signaling Technology, 1:1000 dilution), t-AKT (Cell Signaling Technology, 1:1000 dilution), IκBα (Santa Cruz: sc-371, 1:1000 dilution), and β-actin (DSHB JLA20, 1:2000 dilution). Membranes were next washed and incubated with peroxidase-conjugated secondary antibodies: goat anti-mouse IgG (Fab specific)-peroxidase antibody, (Sigma-Aldrich, A9917, 1:5000 dilution), goat anti-rabbit IgG (whole molecule)-peroxidase antibody, (Sigma-Aldrich, A0545, 1:5000 dilution) in TBST with 1% skim milk for 1 h. Membranes were developed with Western Lightning Ultra Chemiluminescence reagent (Cat#: NEL113001EA, Perkin Elmer), and imaged with an ImageQuant LAS 4000 (GE Healthcare).

## RNA extraction and quantitative PCR

Bone tissue samples were harvested from mice, snap-frozen in liquid nitrogen, and ground into powder using a hammer. The powdered bone tissue samples were transferred into RNAse-free microcentrifuge tubes and RNA was extracted using TRIzol reagent (Cat#. 15596026, Invitrogen), followed by purification using the PureLink RNA mini kit (Cat#. 12183020, Invitrogen). For cell cultures, media was removed before the addition of the TRIzol reagent. Following extraction, RNA samples were treated with RNase-free DNase I (Cat#. 18068015, Invitrogen). For the first-strand synthesis of cDNA, 1–2 μg of total RNA was reverse transcribed using SensiFAST™ cDNA Synthesis Kit (Cat#. BIO-65054, Meridian Bioscience) according to the manufacturer's protocol. Quantitative PCR was performed using SensiMix™ II Probe Kit (Cat#. BIO-83005) with Universal ProbeLibrary (Roche Diagnostics), and the CFX96 Touch System (Bio-Rad). Relative fold expression was normalized to β-actin (ACTB; TaqMan), hydroxymethylbilane synthase (HMBS), and hypoxanthine-guanine phosphoribosyltransferase (HPRT) housekeeping control using primer sequences listed below.

## Oligonucleotides for qPCR

Primer: SLC37A2 isoform 2 Forward: tgctgcatcatgctgatctt; Reverse: cccattctggccaatgtagt

Primer: RUNX2 Forward: cgtgtcagcaaagcttctttt; Reverse: ggctcacgtcgctcatct

Primer: MMP9 Forward: agacgacatagacggcatcc; Reverse: tcggctgtggttcagttgt

Primer: MMP13 Forward: cagtctccgaggagaaactatgat; Reverse: ggactttgtcaaaaagagctcag; Primer: MMP14 Forward: gagaacttcgtgttgcctga, Reverse: ctttgtgggtgaccctgact

Primer: WNT16 Forward: catgaatctacacaacaacgaagc, Reverse: ttttccagcaggttttcaca;

Primer: WNT10b Forward: aatgcggatccacaacaac, Reverse: ctccaacaggtcttgaattgg

Primer: TGFβ1 Forward: tggagcaacatgtggaactc, Reverse: gtcagcagccggttacca

Primer: HPRT Forward: ggagcggtagcacctcct, Reverse: ctggttcatcatcgctaatcac

Primer: ACP5 Forward: cgtctctgcacagattgcat, Reverse: aagcgcaaacggtagtaagg

Primer: CTSK Forward: cgaaaagagcctagcgaaca, Reverse: tgggtagcagcagaaacttg

Primer: HMBS Forward: cagtgatgaaagatgggcaac, Reverse: aacagggacctggatggtg

Primer: ALP Forward: cggatcctgaccaaaaacc, Reverse: tcatga tgtccgtggtcaat

Primer: SOST Forward: atcccagggcttggagagta, Reverse: ccggttca tggtctggtt

Primer: SPHK1 Forward: tgtgaaccactatgctgggta, Reverse: gcagccca gaagcagtgt

Primer: SEMA4D Forward: aagtgggtgcgctacaatg, Reverse: gggcctc actgtcgatacac

Primer: BMP6 Forward: acatggtcatgagctttgtga, Reverse: gtcgttg atgtggggagaac

Primer: RANK Forward: gtgctgctcgttccactg, Reverse: agatgct cataatgcctctcct

Primer: RANKL Forward: tgaagacacactacctgactcctg, Reverse: ccca caatgtgttgcagttc

Primer: OPG Forward: gtttcccgaggaccacaat, Reverse: ccattcaa tgatgtccaggag

## Immunofluorescence confocal microscopy

To preserve the integrity of the tubular SL network, rapid cellular fixation under precise temperature control and microtubule stabilization were performed according to the methods described in ref. [67]. Briefly, osteoclasts were placed on a 37 °C heating block prior to the addition of freshly made, pre-warmed (37 °C) 8% PFA in 2x microtubule-stabilizing buffer (MTSB, 160 mM PIPES pH 6.8, 10 mM EGTA and 2 mM MgCl$_2$) directly to the cell culture media in a 1:1 ratio (final concentration of 4% PFA in 1x MTSB). The cells were then immediately returned to a 37 °C incubator for 15 min. Following fixation, cells were rinsed gently thrice in pre-warmed PBS to remove residual fixative and stored in PBS at 4 °C until required. For immunostaining, osteoclasts were permeabilized with 1% saponin in PBS for 5 min at RT and then washed twice with PBS. Cells were blocked in 3% BSA/PBS for 30 min at RT. Cells were incubated with indicated primary antibodies: ARL8A/B (H-8, Santa Cruz, sc-398635, 1:200 dilution), LAMP-2 (DSHB GL2A7, 1:100 dilution), Rab7 (D95F2, Cell Signaling 9367, 1:100 dilution), Cathepsin K (182-12G5, Millipore, MAB3324, 1:300 dilution), GM130 (35/GM130, BD Transduction Laboratories, 610822, 1:100 dilution), PDI (1D3, ENZO Life Sciences, ADI-SPA-891-F, 1:200 dilution), Vps35 (B-5, Santa Cruz, sc-374372,1:300 dilution) in 0.2% BSA/PBS containing 0.1% saponin overnight at 4 °C. Cells were washed four times with 0.2% BSA/PBS and incubated with secondary antibodies: Alexa Fluor 488 goat anti-mouse IgG (Thermo Fisher Scientific, A-11029, 1:500 dilution); Alexa Fluor 568 goat anti-mouse IgG (Thermo Fisher Scientific, A-11031, 1:500 dilution); Alexa Fluor 647 goat anti-mouse IgG (Thermo Fisher Scientific, A-21236, 1:500 dilution); Alexa Fluor 488 goat anti-rabbit IgG (Thermo Fisher Scientific, A-11034, 1:500 dilution); Alexa Fluor 568 goat anti-rabbit IgG (Thermo Fisher Scientific, A-11036, 1:500 dilution); Alexa Fluor 647 goat anti-rabbit IgG (Thermo Fisher Scientific, A-21245, 1:500 dilution); Alexa Fluor 555 donkey anti-rat, IgG (Thermo Fisher Scientific, A-48270, 1:500 dilution) and Hoechst 33258 (Thermo Fisher Scientific, 1/10,000 dilution) diluted in 0.2% BSA/PBS containing 0.1% saponin for 1 h at RT. Finally, cells were washed three times with PBS and samples were mounted onto glass slides using ProLong Glass Antifade mountant (Invitrogen). Cells were imaged using the NIKON A1R in a Ti-E inverted motorized confocal microscope equipped with Nikon PlanApo ×20 (air), Nikon PlanApo ×60 WI IR, 1.27NA (oil), and Nikon PlanApo ×100, 1.45 NA (oil) immersion lenses.

Immortalized human embryonic kidney cells (HEK293, CRL-1573, ATCC) were maintained in 10% FBS in Dulbecco's modified eagle medium (Life Technologies) supplemented with 2 mM L-glutamine (Life Technologies) and transfected with Lipofectamine 2000 (Life Technologies) according to the manufacturer's instructions. HEK293 cells expressing [emGFP-]Slc37a2 were fixed and immunostained with primary antibodies against CD63/LAMP3 (DSHB H5C6, 1:100 dilution) and GM130 (35/GM130, BD Transduction Laboratories, 610822, 1:100

dilution) according to the methods described above. In some instances, [emGFP-]Slc37a2 expressing cells were labeled with LysoTracker[TM] Red DND-99 (Thermo Fisher Scientific, 1:1000 dilution) probes or co-transfected with late-endosomal/lysosomal markers ([mRFP-]Rab7 or LAMP1[-RFP]) and imaged live using the NIKON A1R in a Ti-E inverted motorized confocal under controlled atmospheric conditions (37 °C and 5% CO$_2$) using an Oko Labs stage top incubator. Frames were captured at 8 s intervals using NIS Elements software (Nikon). Images were processed and channels pseudo-colored using ImageJ (Fiji) software[68].

## Microinjection

Primary BMMs were cultured directly onto ibidi 35 mm glass bottom culture dishes (Cat# 81218-200, ibidi) under osteoclastogenic condition (10 ng/ml RANKL and 25 ng/ml M-CSF) for a period of 4 days. Nuclear microinjection of plasmid vectors including [mCherry-]Slc37a2 isoform 1, [emGFP-]Slc37a2 isoform 2, [RFP-]LAMP1, Cortactin[-dtTomato], [mCherry-]VAMP7, [mCherry-]Rab38), and [mCherry-]Rab5 (all 1 µg/µl in H$_2$O) was performed on an Olympus IMT-2 inverted microscope, using Femtotip II microinjection capillaries (Cat#. 5242957.000, Eppendorf) combined with the Eppendorf electronic microinjector FemtoJet 4x (Cat#. 5253000033, Eppendorf) and InjectMan®4 micromanipulator (Cat#. 5192000035, Eppendorf) using external continuous pressure. After microinjection, cell media was refreshed and the osteoclasts were maintained at 37 °C and 5% CO$_2$ overnight before live cellular imaging the following day.

## Live cell imaging

Primary BMMs were cultured directly onto ibidi 35 mm glass bottom culture dishes under osteoclastogenic conditions (10 ng/ml RANKL and 25 ng/ml M-CSF) for a period of 4–5 days. For live imaging of osteoclasts on bone, similar conditions as above were employed to culture primary BMMs directly on bovine cortical bone slices (Bone-slices.com) for 8–10 days, before the bone slices were transferred to a glass bottom culture dish for imaging. Prior to imaging, osteoclasts were incubated with live imaging compatible fluorescent probes: SIR-Actin (Cytoskeleton, Inc.), LysoTracker[TM] Red DND-99 (Thermo Fisher Scientific), LysoTracker[TM] Green DND-26 (Thermo Fisher Scientific), DQ[TM] Red BSA (Thermo Fisher Scientific), OsteoSense 680 fluorescent probe (Perkin Elmer), Wheat Germ Agglutinin (Thermo Fisher Scientific), Alexa Fluor[TM] 488 conjugate (Thermo Fisher Scientific), Magic Red Cathepsin K Assay (ImmunoChemistry Technologies LLC), Mito-Tracker[TM] Red CMXRos (Thermo Fisher), Hoechst 33258 Pentahydrate (bis-Benzimide) (Thermo Fisher Scientific), according to manufacturer's instructions.

Live osteoclasts treated with fluorescent live probes or micro-injected with plasmids of interest were imaged using the NIKON A1R in a Ti-E inverted motorized confocal under controlled atmospheric conditions (37 °C and 5% CO$_2$) using an Oko Labs stage top incubator. Frames were captured at indicated time points using NIS Elements software (Nikon). Images were processed and channels pseudo-colored using ImageJ (Fiji) software[68]. For sucrose-mediated enlargement of lysosomes, BMM-derived osteoclasts from WT and Slc37a2[KO] mice were incubated with 30 mM sucrose in complete α-MEM for 24 h after which cells were incubated with LysoTracker[TM] Red DND-99 dye for 30 min to label sucrosomes. Sucrosome resolution was initiated by the addition of 0.5 mg/ml, of invertase (S. cerevisiae) for 1 h, and then images were captured continuously by time-lapse confocal microscopy for a period of 10 min.

## Constructs

[mCherry-]Slc37a2 iso1 and [emGFP-]Slc37a2 iso2 were generated using Genscript Express Cloning SC1691 cloning service. Briefly, [mCherry-]Slc37a2 iso1 was designed by cloning the full-length coding sequence (CDS) of Slc37a2 variant 1 (NCBI reference sequence: NM_001145960.1) into the

pcDNA3.1+ vector. The initial ATG start codon was removed and replaced with a mCherry sequence, followed by a 21 bp linker sequence (5'-TCCGGACTCAGATCTCGAGCG-3') downstream. emGFP-Slc37a2 iso2 was generated by cloning the full length CDS of Slc37a2 variant 2 (NCBI reference sequence: NM_020258.4) into the pcDNA3.1+ vector. Similarly, the initial ATG start codon was removed and replaced with an emGFP sequence, followed by a 21 bp linker sequence (5'-TCCGGACTCAGATCTCGAGCG-3') downstream.

To create the entry clone required for gateway cloning into the lentiviral destination vector, SalI, and NotI sites were inserted into emGFP-Slc37a2 iso2 using the following primers (5'-TAAGCAGTCGA-CATGGTGAGCAAGGGCGAG-3'; 5'-TGCTTAGCGGCCGCTCAAA TTTGTTTGTACCCACTGC-3'), then ligated into the pENTR1A no ccDB (w48-1) vector (a gift from Eric Campeau and Paul Kaufman, Addgene plasmid # 17398; RRID:Addgene_17398) using the Quick Ligation™ kit (New England BioLabs) according to manufacturer's instructions. The pLenti PGK Hygro emGFP-Slc37a2 iso2 vector (referred to as pLenti emGFP-Slc37a2) was constructed by recombining pENTR1a emGFP-Slc37a2 iso 2 with the pLenti PGK Hygro DEST (w530-1) vector (a gift from Eric Campeau and Paul Kaufman, Addgene plasmid # 19066; RRID:Addgene_19066) using Gateway LR Clonase II enzyme mix (Thermo Fisher) as per manufacturer's instructions.

### Lentiviral production and transduction of primary bone macrophages
To produce lentivirus, pLenti emGFP-Slc37a2, and pHIV-PV-VSVG (kind gift from Dr. Scott A. Fisher, The University of Western Australia) packaging plasmid were co-transfected into HEK293FT cells (R70007, Thermo Fisher) using TransIT-Lenti transfection reagent (Mirus) according to manufacturer's instructions. 18 h post-transfection, media containing the transfection reagent was removed and replaced with fresh, high-BSA DMEM (Gibco) containing 10% FBS (Gibco) and 0.1 g/L bovine serum albumin (Sigma-Aldrich). Culture media containing lentivirus were collected 48–72 h post-transfection and concentrated using a lentivirus concentration solution (Origene) according to the manufacturer's instructions. The lentiviral pellet obtained post concentration was resuspended in cold, sterile PBS, aliquoted, and stored at −80 °C until required.

Prior to transduction, lentiviral particles were titrated using the qPCR lentivirus titration kit (Applied Biological Materials Inc.) according to the manufacturer's instructions. BMMs were transduced with lentiviral particles in the presence of 20 μg/ml protamine sulfate (Sigma-Aldrich) 24 h after the first RANKL (10 ng/ml, R&D Systems) stimulation of BMMs.

### Bone resorption and extracellular acidification assays
For bone resorption assays, primary BMMs were cultured directly on devitalized bovine cortical bone slices (Boneslices.com) for 10–12 days. Osteoclasts were visualized by incubating with TRAP stain at 37 °C for 30 min. Cells were then washed with PBS and imaged using a standard, upright light microscope. To expose the underlying resorption pits, cells were mechanically removed by gentle brushing of the bone surface with a wet cotton bud. Slices were then washed with 70% ethanol, rinsed in PBS, and left to air-dry overnight. For detection of resorption pits using light microscopy, surfaces of the bone slices were stained with 1% toluidine blue for 30 s, excess toluidine blue was removed by blotting with tissue paper and then imaged using the Nikon Eclipse T200 reflective microscope (NIKON) at ×10 magnification. Images were captured in JPG format using Nikon digital sight DS-5MC and NIS BR Elements 3.2 software. Qualitative analysis of the resorption areas was performed using ImageJ (Fiji) software. To detect resorption pits using SEM, resorbed bone slices devoid of cells were mounted onto aluminum studs and imaged using the TM4000 Plus tabletop SEM (Hitachi). For confocal microscopy, cleaned bone slices were incubated in 0.2% BSA−PBS containing rabbit anti-collagen I

antibody (Abcam, ab34710, 1:300 dilution) for 2 h at RT. Slices were washed in PBS and subsequently incubated with Alexa Fluor 568 goat anti-rabbit IgG (Thermo Fisher Scientific, A-11034, 1:500 dilution) and IVISense Osteo 680 fluorescent probe (Perkin Elmer) for 1 h at RT. Bone slices were washed in PBS before mounting onto glass slides with ProLong™ Diamond antifade mountant and left to cure overnight. The samples were imaged using ×100 oil immersion lenses. Image stacks were opened in Fiji and maximally projected after using Temporal-Color Code to differentially pseudo-color each z depth from the start to the end of the image stack.

For the assessment of extracellular acidification activity, osteoclasts were differentiated on bone-mimicking substrates (Osteo Assay Stripwell microplate, Corning) for 7 days after which Von Kossa staining was performed to visualize the remaining mineralized matrix. Briefly, osteoclasts were removed by incubating in a 25% bleach solution in distilled water for 5 min at RT. Wells were rinsed twice with distilled water before 100 μl 1.5% silver nitrate (Sigma-Aldrich) was added to each well and incubated in the dark for 5 min at RT. The silver nitrate solution was removed and the wells were rinsed well with several changes of distilled water. An equal volume of freshly prepared 0.5% hydroquinone was then added to each well. Precipitation reaction resulted in the remnant mineral turning dark brown/black upon the addition of hydroquinone. After a 1 min incubation, the hydroquinone solution was removed and the wells were washed gently with distilled water. Demineralized zones appear as white clearings against the black mineral background and correlate with the degree of extracellular acidification by osteoclasts.

### Mineralization and Alkaline Phosphatase (ALP) assay
Primary mouse osteoblasts were isolated from the long bones and calvarial bone of WT and Slc37a2$^{-/-}$ mice. Mineralization assays were conducted using $4 \times 10^4$ cells/well (24-well plate) in complete α-MEM supplemented with either 50 μg/mL L-ascorbic acid or osteogenic media, i.e. complete α-MEM with 50 μg/mL L-ascorbic acid, 2 mM β-glycerophosphate (Sigma-Aldrich). Culture medium was replaced every other day and fixed after 21 days using 4% paraformaldehyde. Bone nodules were stained using Alizarin Red S solution (Sigma-Aldrich; 130-22-3). ALP activity was visualized using Leukocyte Alkaline Phosphatase Kit (Sigma-Aldrich; 86R-1KT).

### Histology and histomorphometry
Left proximal tibia and distal femur samples from sex and age-matched wild-type and knockout littermates were processed using a Leica TP1020 processor (Leica Biosystems) in preparation for methyl methacrylate (MMA) embedment according to standard protocol. Samples were sectioned using a Leica Biosystems RM2255 automated microtome at a thickness of 5 μm. Sections were stained for TRAP, Mason's trichrome, and Von Kossa, according to standard protocols. Sections were scanned using a Leica Biosystems Aperio ScanScope. Histomorphometric analysis was performed using BioQuant Osteo, version 13.2.6 (Bioquant Image Analysis, Nashville, TN). For dynamic bone histomorphometry, 11-week-old mice were injected (intraperitoneally) with 5 mg/kg fluorochrome-labeled calcein green (Sigma-Aldrich) twice over the 6-day intervals (i.e. on day 0 and then day 5). Mice were sacrified 2-days after the second injection. Femora and tibia bone tissue were removed, fixed, embedded in SCEM-L1, cryo-sectioned, and slides imaged by confocal microscopy. The mineral apposition rate (MAR) in micrometers per day was calculated from fluorochrome double calcein labels at the periosteal surfaces using NIS Elements software (Nikon).

### Skeletal staining
For, whole mount skeletal staining, euthanized mice were skinned, eviscerated, and fixed in 90% ethanol for 7 days and prepared for

whole mount staining with Alcian blue and Alizarin red S. Briefly, Alcian blue staining (5:1 mixture of 96% ethanol and glacial acetic acid with 0.01% Alcian blue) was performed for 3 days at room temperature. Samples were then postfixed in 96% ethanol (6 h) and cleared in 2% KOH. For Alizarin red staining, samples were incubated for 24 h in Alizarin red S (0.003% Alizarin red S in 1% KOH) and washed in 1% KOH solution three times for 5 hrs. Specimens were then placed in 20% glycerine–1% KOH for 24 h.

### Serum calcium, TRAP5b, CTX-I, and PTH analysis
Serum concentrations of calcium, tartrate-resistant acid phosphatase isoform 5b (TRAP5b), C-terminal telopeptides of type I collagen (CTX-I), and mouse parathyroid hormone (PTH) were measured using the Calcium Assay Kit (Colorimetric) (Cat#. ab102505, Abcam), RatTRAP™ (TRAcP 5b) ELISA (Cat#. SB-TR102, Immunodiagnostics Systems), RatLaps™ (CTX-I) ELISA (Cat#. AC-06F1, Immunodiagnostics Systems), and PTH ELISA Kit (Cat#. CSB E08690m, Cusabio) according to manufacturer's protocols.

### Micro-computed tomography (μCT)
Femurs, tibia, vertebrae, and skulls were scanned at a voxel size of 8.89 μm using Skyscan1176 (Bruker, Kontch, Belgium) Version1.1 (build 10) at X-ray source voltage 50 kVp and current 500 μA. Scans were reconstructed with NRecon version 1.6.10.4 (64 bit) and then analyzed with Bruker CTAn (CT-analyser) version 1.14.4.1. 300 slices below the growth plate were analyzed for the trabecular bone and 100 slices for the cortex. 3D models of tibias and femurs were visualized using Skyscan CTvox software version 3.0.0r1114 (Bruker, Kontich, Belgium).

Analysis was performed on trabecular bone volume, trabecular thickness, trabecular separation, and trabecular number. Cortical bone analysis was performed on cortical thickness.

High-resolution μCT was performed on 12-week-old femurs to assess periosteal bone lesions using a Carl Zeiss X-ray Microscope Versa 520. In this instance, bone samples were scanned at 3 μm resolution using Versa 520 v10.6.2005.12038, then reconstructed with XMReconstructor v10.7.36.79.13921 and finally 3D models assembled with TXM3DViewer version.

### OVX-Induced bone loss
Eight-week-old WT and *Slc37a2*^KO female mice were ovariectomized or sham-operated. After 7 weeks (i.e. 15 weeks of age), mice were sacrificed and subjected to μCT analysis as described above.

### Three-point bending test
Femurs were collected from 12-week-old WT and *Slc37a2*^KO male mice in saline solution and stored at −80 °C. Bones were slowly thawed at 4 °C and then warmed to room temperature. Femur lengths and diameters were measured using calipers. The three-point bending test was performed using an Instron 5566 testing machine (Instron Pty Ltd) with a 100 N load cell and a custom-designed bending rig with a 10 mm span at the midshaft of femurs with a displacement rate of 1 mm/min until the bone fractured with the force and displacement acquired digitally. The ultimate load (N) and stiffness (slope of the linear part of the curve, representing the elastic deformation in N/mm) of the midshaft were calculated according to the corresponding load and femur diameter.

### Metabolite extraction
BMM-derived osteoclasts from WT and *Slc37a2* ^KO mice were cultured on six-well plates, washed with PBS, and then immediately quenched with liquid nitrogen. For metabolite extraction, 400 μl of ice cold 9:1 of methanol:chloroform mixture (containing 10 μM of D-Sorbitol-13C6 and L-Valine-13C5,15N as internal standards) was added to each well, and cells were harvested by the rapid scraping

of cells using a cell scraper (Sarstedt) and collected into 2 ml Eppendorf tubes. 100 μl of chloroform (Sigma Aldrich) was added to each tube and samples were vortexed vigorously for 15 s before shaking on an ice block for 15 min. Samples were then centrifuged at 16,000×*g* at 4 °C for 3 min. The polar phase was collected into fresh tubes and 150 μl of $H_2O$ (GC-MC grade, Sigma Aldrich) was added to each sample and vortex at high speed for 1 min. Samples were centrifuged and 300 μl of the aqueous phase was collected into GC-MS glass vials and dried in a SpeedVac (Labconco) at RT overnight. Data acquisition of metabolite profiles was performed as described by Dias et al.[69] with slight modifications. Polar metabolite extracts were derivatized by methoxyamination and trimethylsilylation using a Gerstel MPS2 XL automated sampling system. During this process, the samples were shaken for 2 h at 37 °C and 650 rpm in 20 μl of MeOX before being incubated for 1 h at 37 °C and 650 rpm in 30 μl of methyl-trimethylsilyl trifluoroacetamide (MSTFA) mixture.

Metabolite profiles were acquired utilizing a 7890A Gas Chromatograph coupled with a 5975C mass selective detector (Agilent Technologies, Santa Clara, CA, USA). 1 μl of the sample was injected onto an Agilent VF- 5msec (J&W, Australia) in splitless mode and inlet temperature of 280 °C. Helium was used as the carrier gas at a flow rate of 1 ml/min. Oven conditions were set at 70 °C starting temperature, held for 1 min, then ramped at 7 °C/min to 325 °C and held for 5 min. Data acquisition was controlled by the Agilent MassHunter software package (B.05.00). Chromatogram processing and metabolite identification were performed utilizing AMDIS software (v 2.73), the in-house Metabolomics Australia mass spectral library, and the integrated NIST14 database (National Institute of Standards and Technology, Gaithersburg, MD, USA). The putative results were normalized by the abundance of internal standards in each sample and corrected for weight.

### Electron microscopy
For transmission electron microscopy long bones from 5-day-old WT and *Slc37a2*^KO littermates were dissected free of soft tissue and fixed with 2.5% glutaraldehyde, 4% paraformaldehyde in 0.01 M cacodylate buffer (pH 7.3) for at least 1 h at room temperature. After the removal of the fixation solution, bones were decalcified for 5 days in 14% EDTA (pH 7.4) containing 0.1% glutaraldehyde. Subsequently, bones were washed, cut into ~1 mm slices just below the growth plate, and postfixed with 1% osmium tetroxide, dehydrated in ethanol, and embedded in Epon 812 (Nissin EM, Tokyo). Ultrathin sections were counterstained with lead citrate and examined with a Philips 410 LS transmission electron microscope (Phillips Inc., Eindhoven, The Netherlands). For assessment of SPION-enriched SL fractions, 6 μl of the F2 fraction was absorbed onto Formvar-coated 150 mesh copper grids (ProSciTech) and imaged as described above.

### APEX electron microscopy
Ascorbate peroxidase (APEX) processing of baby hamster kidney (BHK; CCL-10, ATCC) cells was co-transfected with ^emGFP-Slc37a2 and APEX-GBP (GFP-binding peptide) was performed according to the methods described in ref. [70]. Briefly, transfected cells were fixed in 2.5% glutaraldehyde, and washed repeatedly in 0.1 M sodium cacodylate buffer prior to the DAB reaction. Cells were incubated with DAB in the presence of $H_2O_2$ for 30 min at RT, washed in 0.1 M sodium cacodylate buffer, post-fixed in 1% osmium tetroxide for 2 min, washed again, then serially dehydrated in increasing percentages of ethanol. Cells were serially infiltrated with LX112 resin in a Pelco Biowave microwave and then polymerized overnight at 60 °C. Ultrathin sections were cut on an ultramicrotome (Leica EM UC6, Leica Microsystems) and imaged using a JEOL1011 electron microscope (JEOL) at 80 kV.

## RNAseq

For RNAseq analyses, femurs and tibias were dissected from three male and female WT and *Slc37a2*[KO] mice followed by whole-bone RNA extraction. Briefly, bones were frozen in liquid nitrogen and crushed before the addition of Trizol with vigorous shaking. Chloroform was added, and the samples were spun down at 12,000 rpm for 30 min at 4 °C. The aqueous phase was collected and mixed with one volume of 70% ethanol. Subsequent RNA extraction steps were performed with the Life Technologies RNA extraction kit (Cat. No. 12183018A). RNA sequencing was performed through the Illumina sequencing service by the Australian Genome Research Facility. Image analysis was performed in real-time by the NovaSeq Control Software (NCS) v1.7.0 and real-time analysis (RTA) v3.4.4, running on the instrument computer. RTA performs real-time base calling on the NovaSeq instrument computer. Illumina bcl2fastq 2.20.0.422 pipeline was used to generate the sequence data. The primary bioinformatics analysis involved demultiplexing and quality control (QC). The data was then processed through an RNA-seq expression analysis workflow, which included alignment, transcript assembly, quantification, and normalization. Analysis of the Top 1000 differentially expressed genes was performed with edgeR (version 3.30.3) using R4.0.0.

## Statistics

Statistical analysis was performed using GraphPad Prism (GraphPad Software 9.0.2, San Diego, CA). For the experimental cohorts, 3–16 mice per genotype per experiment were used (=biological replicates) as indicated in the figures/figure legends for each experiment. For mouse experiments, age, and sex-matched wildtype littermates were used as controls. All experiments were performed at least twice with the indicated number of replicates. Sample sizes were based on experience and experimental complexity but no methods were used to determine the normal distribution of the samples. Tests between two groups were carried out using unpaired, two-tailed Student's *t*-test. Multiple comparisons were assessed using nested one-way ANOVA with Tukey's posthoc analysis. Data are presented as mean ± standard deviation (SD).

## Reporting summary

Further information on research design is available in the Nature Portfolio Reporting Summary linked to this article.

## Data availability

The proteomic data that support the findings of this study have been deposited to the ProteomeXchange Consortium via the PRIDE partner repository under accession numbers PXD037006, PXD037813, and PXD037910. The RNAseq data produced in this study were deposited to the public database (GSE219216). All data supporting the findings of this study are available within the article and its Supplementary Information files. Source data are provided with this paper.

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

## Acknowledgements

We thank Rob Day and Alex Haynes (Royal Perth Hospital, WA) for their technical assistance with the biomechanical testing studies, Lisa Griffiths (PathWest, WA) for performing electron microscopy on bone tissue, Daniel Yagoub for proteomics assistance, James Rae for assistance with electron microscopy of cultured cells and Natalie Sims (St. Vincent's Institute, VIC) for insightful discussions. We are also indebted to Michael S. Marks (University of Pennsylvania, USA) and Alistair N. Hume (The University of Nottingham, UK) and Roland Baron (Harvard School of Dental Medicine, USA) for the provision of plasmids. This work was supported by NHMRC Project funding APP1143921 to N.J.P., R.D., and J.X., NHMRC grants APP1140064 and APP1150083 and fellowship APP1156489 to R.G.P., an NHMRC grant APP2003629 and Department of Health Western Australia Merit Award 1186046 to B.M., an Arthritis Australia and HJ & GJ Mackenzie Grant (N.J.P. and P.Y.N.), and a Faculty of Health and Medical Sciences Research Grant Scheme (SE Ohman Medical Research Fund to N.J.P. and P.Y.N.). A.B.P.R. is supported by Australian Government Research Training Program Scholarship. The authors acknowledge the facilities, and the scientific and technical assistance of Microscopy Australia at the Centre for Microscopy, Characterization & Analysis, The University of Western Australia, a facility funded by the University, State, and Commonwealth Government and of the Microscopy Australia Research Facility at the Centre for Microscopy and Microanalysis at The University of Queensland. N.J.P. and K.S. are supported by COST Action GEMSTONE CA18139 (European Cooperation in Science and Technology).

## Author contributions

P.Y.N. performed the experiments, analyzed the data contributed to manuscript drafting and editing. A.B.P.R. performed experiments, analyzed data contributed to manuscript drafting and figures, Q.G. performed micro-CT analyses, cryosectioning, and biomechanical testing. B.H.M. performed GWAS analysis and contributed figures, J.W.Y.T. contributed in vitro experiments and metabolomics, E.L.B. performed micro-CT and histomorphometric analyses, S.S. contributed to the establishment of microinjection experiments, K.C. and J.Y. assisted with micro-CT analyses and OVX-studies, L.A. assisted with proteomic analyses, M.B. contributed to the metabolomic analyses, E.T.T.T.N. assisted with histomorphometric analyses, J.K. analyzed the RNAseq data, J.M.P. performed histopathological analyses and critically reviewed the manuscript, K.S. analyzed the in vitro bone resorption data and edited and critically reviewed the manuscript, R.D.T. and J.X. financially supported and critically reviewed the manuscript, R.G.P. performed APEX work, edited and critically reviewed the manuscript, H.T. offered critical suggestions to the project design and manuscript drafting, N.J.P. conceptualized the project, designed all experiments, wrote the manuscript and financially supported the project.

## Competing interests

The authors declare no competing interests.
