## [Peer Review File · Nature Communications]

Sugar transporter Slc37a2 regulates bone metabolism via a dynamic tubular lysosomal network in osteoclastsREVIEWER COMMENTS

Reviewer #1 (Remarks to the Author):

This manuscript unveils a new pathway, the Slc37a2 sugar transporter, relevant for osteoclast function, that contributes to the ruffled border formation. In vivo studies on a global Slc37a2 knockout mouse model show a high bone mass phenotype, which not only involves the osteoclasts but also the osteoclast-osteoblast crosstalk. The manuscript is well written and organised and, in general, shows well documented results. However, there are some major concerns that need to be addressed as indicated in the specific comments.

Major comments

1. Lines 101-108: The osteoclastogenesis timing, especially regarding the pre-osteoclast to mature osteoclast switch, should be shown experimentally in the culture conditions used for the study. Specifically, it is claimed that the appearance of the $\alpha\text{V}\beta\text{3}$ integrin defines the switch from pre-osteoclasts to mature osteoclasts and that at this stage >90% of the cells are $\alpha\text{V}\beta\text{3}$ integrin positive. This time-course should be shown to demonstrate that indeed the 3 days RANKL treatment used in the study matches with the pre-osteoclast status indicated in line 103.
2. Line 113: Last right panel of Fig 1A must be described in the text.
3. Line 148: Figure 1D should be described better as it is quite difficult to go through the results in comparison to Fig. 1C. Perhaps, showing in Fig. 1D the Fig. 1C identified genes with a green box, while the Slc37a2 remains marked with the red box, may help the reader to interpret this figure.
4. In Fig. 1C, Slc37a2 is not the most enriched gene in the lysosomal compartment. This should be remarked in the results section.
5. The order of Fig. 2B, 2C, 2D and 2A should be changed and described from A to D, with A being the WB and D being the schematic representation of the Slc37a2 protein.
6. Current Fig. 2A: The position of the spliced C-terminus amino acids in the two Slc37a2 isoforms should be indicated in the figure.
7. Fluorescence figures and videos: My major concern in this paper is that while the Slc37a2 tubule formation is promptly evident in these documentations, the fusion of tubules with the ruffled border membrane is unclear. I do not want to argue here that this is not the case, but this set of results should be demonstrated uniquely. In the picture, it is not clear to me while single staining is shown in black and white, while only the merge figures are shown in colour. Could you please explain the rationale or, even better, use always colour figures?
8. In some figures, for instance 2F and 2L, as well as in video S4, what is clear is that the tubules reach the cell edge, but the fusion with the membrane is not evident. Please provide convincing demonstration that fusion occurs at this level.
9. Line 224-225: That "Slc37a2+ tubular lysosomal network was highly sensitive to chemical and cold fixation (i.e. 4% PFA)" is not shown. Please, provide a figure that shows this result.
10. Line 222: In video S2, some overlay between Slc37a2 and Rab5 marking endosomes is apparent in orange, while in line 222 it is stated that Slc37a2+ tubules and endosomes are morphologically and dynamically distinct.
11. Lines 237-239: The conclusion that "that Slc37a2 occupies a network of acidified cathepsin-containing tubular LROs that fuse with the bone-apposed plasmalemma" is not fully supported by the data (see above).

12. Lines 311-314: The conclusion that impaired bone resorption is compensated by increased expression of proteases is quite naïve. This increase cannot compensate bone resorption because without bone demineralization by osteoclasts, no enzyme has access to the organic matrix components and bone resorption remains impaired. This is exactly what occurs in osteoclast-rich osteopetrosis.

13. The cartoon of Fig. 7D is too small and cannot be observed easily by the reader.

14. Lines 414-417, lines 424-426 and lines 468-469: These conclusions are based on the concept that Slc37a2⁺ tubules fuse with the plasma membrane. Please, see my concerns above.

15. Lines 492-494: Again, although plausible, this result is not fully demonstrated.

16. Figure 3D-K and lines 495-505: Such a high impairment of osteoclast erosive activity (see Fig 6H) should induce a typical osteopetrotic phenotype in which, given the lack of bone remodelling, bone strength should be reduced rather than increased. How do you explain that stiffness (Fig. 3J) and ultimate force (Fig. 3K) are higher in the KO mice? In my opinion, the discussion in lines 495-505 should be revised as it appears rather speculative and confused.

17. Lines 498-499: enhanced osteoclastogenesis when bone resorption is abolished can be due to hypocalcaemia that stimulates PTH release, which in turn induces RANKL expression. What is the serum calcium and PTH concentration in the Slc37a2 KO mice?

Minor comments

1. In general, the use of non-standard abbreviations should be limited as much as possible to allow unexpert readers to easily understand the concepts. For instance, SLs, LROs, PNS, PFA ect. confuse readers especially if their cell biology background is not strong.

2. Line 65: Tartrate-resistant acid phosphatase should be mentioned in this sentence as an important osteoclast endo-lysosomal protein.

3. Line 255: The term mammography is wrong.

4. Figures should not be mentioned in the discussion except the summary graphical representation of Fig. 8.

Reviewer #2 (Remarks to the Author):

The manuscript by Ng et al presents an interesting set of high-quality data, identifying putative sugar transporter Slc37a2 as a new regulator of bone metabolism. Slc37a2 is enriched in purified secretory lysosomes and the global Slc37a2 KO in mice produces pronounced bone defects. Although the majority of the data is well controlled and high quality, some of the controls are missing and the data on localization and cellular function of Slc37a2 are overinterpreted. Specifically, the association of Slc37a2 with the "dynamic tubular lysosomal network" is only shown for the overexpressed tagged protein. This interesting observation should be confirmed for the endogenous protein or discussed and severely toned down.

Comments:

- The exact intracellular localization of the Slc37a2 in osteoclasts is essential for this study. A previous study (Pan et al, 2011) identify Slc37a2 as ER-resident protein and staining in Fig 2C could not discriminate between ER and endolysosomal patterns. Control for IF specificity of Slc37a2 antibodies (staining of osteoclasts isolated from Slc37a2 KO animals) and quantified co-

localization of the endogenous Slc37a2 with ER (sec61), Golgi (GM130) and endolysosomal markers is needed.

- Molecular weight marker should be added in Fig2A blots. It is peculiar that Slc37a2 signal is not diminished in the UB lane. This behavior is different from the behavior of LAMP2. If the Slc37a2 signal is not reduced after SL isolation, where is the remaining Slc37a2 localized?
- FP-Slc37a2 is likely to be severely overexpressed under CMV promoter and its co-localization with lysosomal marker could be a result of the overexpression. Also, the “dynamic tubular lysosomal network” could be an artifact of overexpression of FP-Slc37a2. In fact, LysoTracker-positive tubules are more prominent in cells overexpressing FP-Slc37a2 (Fig 2G) as compared to the normal cells (Fig 7B, extended Fig 2A). Immuno-EM with antibodies to the endogenous Slc37a2 should be helpful to resolve this issue. Is FP-Slc37a2 functional? Could it rescue Slc37a2 KO cells or animals?
- In the ovariectomy experiment, the variation in the femoral BV/TV values of KO group seems to be masking a decline in bone mass. However, a statistically significant decline in trabecular thickness is still observed in femoral cancellous bone. It seems these KO mice start with very high bone mass and lose some bone in the femurs (albeit at levels lower compared to controls). It may be more appropriate to call this partial protection in femurs. What happens to cortical thickness in these mice in response to OVX? Is Ctx increased in response to OVX in KO mice?
- The presentation of data on the role of Slc37a2 in the export of monosaccharide sugars from SL (Figure 7D) should be improved. Scale bars and quantification should be included. How invertase is getting into secretory lysosomes?
- The last sentence of the discussion mentions that KO mice are protected from age-related bone loss, but this data is not included in the manuscript. It would be great to see this data or have this sentence corrected.
- In the cartoon working model in Fig 8, osteoclasts seem to be producing more SP1 and Tgf β to stimulate the increase in osteoblasts. Increases in mRNA levels of these factors are observed in whole bone preparations of “ femurs” (Fig 5P) but their cellular source is not clear. Tgf β is expressed by numerous cell types in the bone niche including osteoblasts. Moreover, as Tgf β that is released from the bone matrix due to resorption is thought to play a role in coupling resorption to the formation, and the KO mice do not have increased resorption (total ctx is the same in both groups). Therefore, it is unclear how changes in Tgf β mRNA levels fit into the cellular phenotype of these mice. Do the authors measure levels of these factors in osteoclast culture media or their mRNA levels in cultured osteoclasts (similar to what has previously been done with Sp1 of Ctsk null osteoclasts)? Are these factors increased due to the increased number of osteoclasts in KOs or is their production/release affected in KO osteoclasts?
- The presence of cartilaginous lesions in femurs is very interesting. Are these lesions observed only in femurs or are they present in the vertebral cortex as well?

Minor comments:

- Please change the % change in cancellous bone mass in line 259 regarding Fig 3E to fold change as opposed to % change.
- Please clarify whether the femur gene expression is whole femur or flushed femur etc. in fig 5 E and P.
- The addition of landmarks or yellow dash lines would help orient readers in the KO image of Fig 4G.
- (i) Evidence that elevated Rankl observed in this model is due to the increased population of osteoblasts as depicted in the cartoon, or (ii) a sentence or two on other possible cellular sources of Rankl in this model would be a nice addition.

RESPONSE TO REVIEWER COMMENTS

Reviewer #1 (Remarks to the Author):

This manuscript unveils a new pathway, the Slc37a2 sugar transporter, relevant for osteoclast function, that contributes to the ruffled border formation. In vivo studies on a global Slc37a2 knockout mouse model show a high bone mass phenotype, which not only involves the osteoclasts but also the osteoclast-osteoblast crosstalk. The manuscript is well written and organised and, in general, shows well documented results. However, there are some major concerns that need to be addressed as indicated in the specific comments.

We thank Reviewer 1 for the kind comments and recommendations. Please find our responses to the comments raised below.

Major comments

1. Lines 101-108: The osteoclastogenesis timing, especially regarding the pre-osteoclast to mature osteoclast switch, should be shown experimentally in the culture conditions used for the study. Specifically, it is claimed that the appearance of the $\alpha_V\beta_3$ integrin defines the switch from pre-osteoclasts to mature osteoclasts and that at this stage >90% of the cells are $\alpha_V\beta_3$ integrin positive. This time-course should be shown to demonstrate that indeed the 3 days RANKL treatment used in the study matches with the pre-osteoclast status indicated in line 103.

Response: We agree with the Reviewer. The osteoclastogenesis time-course of the SPION pulse-chase analysis $\alpha_V\beta_3$ integrin (IntegriSense⁶⁴⁵) positive osteoclasts has now been included in the revised Figure 1a. In keeping with our original statement, quantitative analyses of the percentage of $\alpha_V\beta_3$ integrin (IntegriSense⁶⁴⁵) positive cells increases exponentially upon RANKL stimulation with ~90% of mononuclear/bi-nucleated pre-osteoclastic cells positive for IntegriSense⁶⁴⁵ at Day 3 and >90% of large multinuclear mature osteoclasts positive for IntegriSense⁶⁴⁵ at Day 5 post-RANKL stimulation (Figure 1b).

2. Line 113: Last right panel of Fig 1A must be described in the text.

Response: Thanks. Fig 1A is now revised Fig. 1c. This information has now been added to the revised text on page 4, line 107-8: “A representative mass spectrum of fraction F2 is depicted in Fig. 1c (far right panel)”.

3. Line 148: Figure 1D should be described better as it is quite difficult to go through the results in comparison to Fig. 1C. Perhaps, showing in Fig. 1D the Fig. 1C identified genes with a green box, while the Slc37a2 remains marked with the red box, may help the reader to interpret this figure.

Response: As recommended we have improved the description of Figure 1d in the figure legend and highlighted Fig1C identified genes with green boxes in the revised Figure 1d (now revised Figure 1e).

4. In Fig. 1C, Slc37a2 is not the most enriched gene in the lysosomal compartment. This should be remarked in the results section.

Response: Thanks. This has now been stated in the results section page 5, Line 156:

“..that although was not the most enriched protein detected on SLs, was enriched 2.35 Log₂FC (P=3.11E⁻⁰²)..”

5. The order of Fig. 2B, 2C, 2D and 2A should be changed and described from A to D, with A being the WB and D being the schematic representation of the Slc37a2 protein.

Response: As recommended, Figure 2 has been reordered. Please note that Figure 2 has been substantially revised to accommodate new endogenous Slc37a2 immunofluorescence data requested by Reviewer 2 (comment #1). The original Western Blot data (Fig. 2B) has been moved to Supplementary Figure 1a.

6. Current Fig. 2A: The position of the spliced C-terminus amino acids in the two Slc37a2 isoforms should be indicated in the figure.

Response: As recommended, the position of the spliced C-terminus is now indicated by a magenta box in the revised Fig. 2c (formerly Fig. 2A) and corresponds to the amino acid sequences presented in Fig. 2d.

7. Fluorescence figures and videos: My major concern in this paper is that while the Slc37a2 tubule formation is promptly evident in these documentations, the fusion of tubules with the ruffled border membrane is unclear. I do not want to argue here that this is not the case, but this set of results should be demonstrated uniquely. In the picture, it is not clear to me while single staining is shown in black and white, while only the merge figures are shown in colour. Could you please explain the rationale or, even better, use always colour figures?

Response: This is a fair comment. However, the bone disc physically obstructs monitoring of tubule fusion directly with the ruffled border membrane as the thickness of the bone disc (~200 μm) goes beyond the axial resolution limits of our optical instrumentation (i.e. the objective lens needs to be within 100 nm of the ruffled border membrane for total internal reflection fluorescence (TIRF) microscopy). Nonetheless, our revised data (Figure 3 c-f) confirms that Slc37a2-positive tubules are concentrated within the ruffled border and that individual tubules undergo transient fusion with the osteoclast plasma membrane (PM) when imaged at the level of the bone surface. As shown below and in revised Figure 3, we initially observe extension of a Slc37a2/lysotracker positive tubule towards the PM (Figure 3e-f) that, in turn contacts the bone-lining PM where it persists for >20 seconds before rapidly retracting (<9 secs). During prolonged contact with the PM, content release (Figure 3f, white arrow) is visible in proximity of the tubule with the PM, indicative of fusion/exocytosis. This was confirmed by the change in fluorescent signal intensities at different stages of the contact/tethering using correlative line scans (Figure 3f). This fusion process is reminiscent of tubular lysosome-to-PM fusion observed in dendritic cells (Chow, A., Toomre, D., Garrett, W. & Mellman, I. *Dendritic cell maturation triggers retrograde MHC class II transport from lysosomes to the plasma membrane. Nature* 418, 988-994 (2002)). As recommended by Reviewer 1, these data have now been presented uniquely and shown in full colour. The revised text (page 7, lines 220-227) and Figure are presented below.

“While the bone disc physically obstructed direct visualization of fusion between ^{emGFP}-Slc37a2⁺ tubules and the ruffled border membrane, transient fusion between tubules and the osteoclast plasma membrane (PM) were captured at the level of the bone surface (Fig. 3 e,f). Here, ^{emGFP}-Slc37a2⁺ tubules extended towards and made prolonged contact with the bone-lining PM (Fig 3e, yellow line) before transiently fusing, as monitored by content release (Fig. 3f, white arrows and line scans), and then rapidly

retracting, a process reminiscent of tubular lysosome-to-PM fusion observed in dendritic cells (Chow, A. et al. Nature, 2002)”.

8. In some figures, for instance 2F and 2L, as well as in video S4, what is clear is that the tubules reach the cell edge, but the fusion with the membrane is not evident. Please provide convincing demonstration that fusion occurs at this level.

Response: Thanks. Please refer to our response above to point 7 raised by Reviewer 1 and revised Figure 3e-f that now demonstrates content release, an indicator of tubule fusion, at the level of the bone-lining plasma membrane.

9. Line 224-225: That “Slc37a2+ tubular lysosomal network was highly sensitive to chemical and cold fixation (i.e. 4% PFA)” is not shown. Please, provide a figure that shows this result.

Response: We apologise for this omission. These data have now been included in Supplementary Figure 4a. As illustrated below, endogenous Slc37a2 occupies an extensive tubulo-vesicular network in mouse bone marrow monocyte derived osteoclasts that is readily observable upon rapid fixation in the presence of microtubule stabilising buffer (MTSB) at 37°C. Comparatively, this intricate network collapses into large Slc37a2 positive puncta when osteoclasts are cold fixed with 4% paraformaldehyde (PFA) at 4°C.

10. Line 222: In video S2, some overlay between Slc37a2 and Rab5 marking endosomes is apparent in orange, while in line 222 it is stated that Slc37a2+ tubules and endosomes are morphologically and dynamically distinct.

Response: To clarify, the orange overlay between $emGFP\text{-Slc37a2}$ and $mCherry\text{-Rab5}$ in video S2 primarily reflects coincidental overlap between the two fluorescent signals rather than genuine spatial overlap of colocalized pixels between these two fluorescent fusion proteins as confirmed by the distinct fluorescent profiles obtained by correlative line scan (see graph below corresponding to scan of white line depicted in Overlay). This was further confirmed by colocalization analyses using Pearson's correlation coefficient (R_r), where $R_r < 0.5$ corresponds to weak to little association between pixels in dual-color fluorescence images. The mean R_r between Slc37a2 and Rab5 = 0.3658 indicative of weak correlation in keeping with our original statement that Slc37a2+ tubules and endosomes are morphologically and dynamically distinct. This position was also confirmed in fixed osteoclasts using an independent marker of endosomes (Vps35) compared to endogenous Slc37a2+ compartments (Revised Figure 2a), in this instance $R_r(\text{Slc37a2:Vps35}) = 0.2120$.

11. Lines 237-239: The conclusion that "that Slc37a2 occupies a network of acidified cathepsin-containing tubular LROs that fuse with the bone-apposed plasmalemma" is not fully supported by the data (see above).

Response: Please refer to our detailed response to point 7 raised above and new data in Figure 3f that now supports our original conclusion.

12. Lines 311-314: The conclusion that impaired bone resorption is compensated by increased expression of proteases is quite naïve. This increase cannot compensate bone resorption because without bone demineralization by osteoclasts, no enzyme has access to the organic matrix components and bone resorption remains impaired. This is exactly what occurs in osteoclast-rich osteopetrosis.

Response: Thanks, this is a valid comment. To clarify, unlike other forms of osteoclast-rich osteopetrosis where osteoclast function is completely abolished, genetic ablation of *Slc37a2* only partially incapacitates osteoclast extracellular acidification and bone resorption function (please refer to Figures 7f, g). In this regard, *Slc37a2*^{KO} osteoclasts still retain some capacity to expose the organic matrix. Nonetheless, we agree with the Reviewer and have removed this statement in the revised text to avoid reader confusion.

13. The cartoon of Fig. 7D is too small and cannot be observed easily by the reader.

Response: The cartoon in Fig. 7D has now been removed to accommodate new data in the revised Figure (now Figure 8). A larger version of the original cartoon is presented in Supplementary Movie 6 for reader ease.

14. Lines 414-417, lines 424-426 and lines 468-469: These conclusions are based on the concept that *Slc37a2*⁺ tubules fuse with the plasma membrane. Please, see my concerns above.

Response: Please refer to our response to point 7 raised by Reviewer 1 above and revised Figure 3e, f that documents fusion of a *Slc37a2*⁺ tubule with the osteoclast plasma membrane.

15. Lines 492-494: Again, although plausible, this result is not fully demonstrated.

Response: Please refer to our response to point 7 raised by Reviewer 1 above and revised Figure 3f in support of our original statement.

16. Figure 3D-K and lines 495-505: Such a high impairment of osteoclast erosive activity (see Fig 6H) should induce a typical osteopetrotic phenotype in which, given the lack of bone remodelling, bone strength should be reduced rather than increased. How do you explain that stiffness (Fig. 3J) and ultimate force (Fig. 3K) are higher in the KO mice?

Response: Thanks, this is an interesting point. Compared with other genetic knockout models of osteopetrosis in which osteoclast differentiation (e.g. RANKL, c-fos) or function is completely ablated (e.g. *TCIRG1*, *Rac1/2*), *Slc37a2*-deficient mice exhibit a milder form of osteopetrosis/ progressive osteosclerosis associated with diminished bone resorptive activity. Bone remodelling is therefore reduced but is not completely abrogated in *Slc37a2*^{KO} mice. Based on our biomechanical testing performed on whole femurs isolated from 12-week old male mice, we attribute the observed increase in central femur bone stiffness (Figure 4j) and strength (ultimate force, Figure 4k) to the presence of increased trabecula and cortical thickness in *Slc37a2*^{KO} mice (Figure 4c-i). These data are consistent with the increased strength and stiffness reported for male cathepsin K-deficient mice that similarly manifest a mild form of osteoclast-rich osteopetrosis (*Li et al., JBMR, 2006; Pennypacker et al., Bone 2009; Gentile et al., Bone, 2014*). However, unlike femurs from cathepsin K KO mice which exhibit significantly lower post yield displacement (PYD) (i.e. less ductile/more brittle bones) (*Li et al., JBMR 2006*), the PYD of *Slc37a2*^{KO} femurs were not statistically different from femurs isolated from wild-type littermates (see figure below). The exact reason for this difference remains unclear and will require detailed analyses of bone material quality indices (both collagen and mineral content) in future studies.

In my opinion, the discussion in lines 495-505 should be revised as it appears rather speculative and confused.

Response: As recommended, we have revised lines 495-505 in the Discussion, now lines 475-479.

17. Lines 498-499: enhanced osteoclastogenesis when bone resorption is abolished can be due to hypocalcaemia that stimulates PTH release, which in turn induces RANKL expression. What is the serum calcium and PTH concentration in the *Slc37a2* KO mice?

Response: This is an excellent point. We have now measured the serum calcium and PTH concentrations in *Slc37a2* KO mice (n=6/group) but found no statistical difference compared with WT littermate controls as shown below. These data are now presented in Supplementary Fig. 7b and we have added the following statement to the Results section page 9, lines 298-301.

*“Because enhanced osteoclastogenesis can result from hypocalcaemia that stimulates PTH release, which, in turn induces RANKL expression, we also checked serum calcium and PTH levels in *Slc37a2*^{KO} mice but found no difference compared with WT controls (Supplementary Fig. 7b)”*.

Minor comments

1. In general, the use of non-standard abbreviations should be limited as much as possible to allow unexpert readers to easily understand the concepts. For instance, SLs, LROs, PNS, PFA etc. confuse readers especially if their cell biology background is not strong.

Response: Thank you. We have revised our use of non-standard abbreviations throughout the manuscript and defined them at first use for reader ease.

2. Line 65: Tartrate-resistant acid phosphatase should be mentioned in this sentence as an important osteoclast endo-lysosomal protein.

Response: Thanks. We agree and have now referenced TRAP (*Baron et al., Am J Pathol, 1986*) as an osteoclast endo-lysosomal protein in this sentence: page 3, line 61.

3. Line 255: The term mammography is wrong. **Response:** Thanks. This has now been corrected.

4. Figures should not be mentioned in the discussion except the summary graphical representation of Fig. 8.

Response: Thank you. We have now removed all references to figures in the Discussion with the exception of the graphical summary (Fig. 8= revised Fig. 9).

Reviewer #2 (Remarks to the Author):

The manuscript by Ng et al presents an interesting set of high-quality data, identifying putative sugar transporter Slc37a2 as a new regulator of bone metabolism. Slc37a2 is enriched in purified secretory lysosomes and the global Slc37a2 KO in mice produces pronounced bone defects. Although the majority of the data is well controlled and high quality, some of the controls are missing and the data on localization and cellular function of Slc37a2 are overinterpreted. Specifically, the association of Slc37a2 with the “dynamic tubular lysosomal network” is only shown for the overexpressed tagged protein. This interesting observation should be confirmed for the endogenous protein or discussed and severely toned down.

We thank Reviewer 2 for the interest in our work and for the valuable suggestions to improve our manuscript.

Comments:

• The exact intracellular localization of the Slc37a2 in osteoclasts is essential for this study. A previous study (Pan et al, 2011) identify Slc37a2 as ER-resident protein and staining in Fig 2C could not discriminate between ER and endolysosomal patterns. Control for IF specificity of Slc37a2 antibodies (staining of osteoclasts isolated from Slc37a2 KO animals) and quantified co-localization of the endogenous Slc37a2 with ER (sec61), Golgi (GM130) and endolysosomal markers is needed.

Response: We agree with the Reviewer. As recommended, we have now included controls for specificity of the Slc37a2 antibody against osteoclasts from *Slc37a2*^{KO} mice. First, we used Western blotting to confirm the specificity of the antibody to detect endogenous Slc37a2 during RANKL-induced differentiation of mouse bone marrow monocytes (BMM) into cathepsin K (Ctsk) expressing osteoclasts (OC). As shown below, the custom Slc37a2 antibody detects a major band (black arrow) and minor band (asterisk) corresponding to the predicted size(s) of glycosylated (~75kDa) and unglycosylated (~55kDa) Slc37a2 in BMM-derived osteoclasts from wild-type (WT) mice but not *Slc37a2*^{KO} mice (Panel a below). Second, we confirmed the specificity of the Slc37a2 antibody by immunofluorescence confocal microscopy. As indicated in Panel b, strong immunofluorescence Slc37a2 signal (green) was concentrated within F-actin rings (red) of a bone-resorbing WT osteoclast whereas this signal was specifically depleted in *Slc37a2*^{KO} osteoclasts. This data now forms Supplementary Figure 1.

In addition, as recommended by Reviewer 2 we performed new experiments comprehensively characterizing the exact intracellular localization of endogenous Slc37a2 in osteoclasts against a panel of well-established endogenous markers for the ER (PDI), Golgi (GM130), endosomes (Vps35) and endolysosomal markers (LAMP2, Rab7 and Arl8). These data are presented below and in the revised

Figure 2a and validate independently the existence of the tubular lysosomal network occupied by Slc37a2 in naïve OCs. As recommended by Reviewer 2, we have also performed quantitative analyses (Figure 2b) to confirm that Slc37a2 is exclusively associated with endo-lysosomal markers ($R_r > 0.6$) but not endosomes (Vps35 $R_r=0.2120$), the ER (PDI $R_r=0.2120$) or the Golgi (GM130 $R_r=0.2120$) in good accord with our live cell data expressing emGFP-Slc37a2 iso2 (see Fig. 2f, g and Supplementary Fig. 2b, c).

Figure 2 a-b

- Molecular weight marker should be added in Fig2A blots. It is peculiar that Slc37a2 signal is not diminished in the UB lane. This behavior is different from the behavior of LAMP2. If the Slc37a2 signal is not reduced after SL isolation, where is the remaining Slc37a2 localized?

Response: The peptide (amino acids CTPPRHHDDPEKEQ) used to raise the custom Slc37a2 antibody corresponds to the cytoplasmic facing loop of mouse Slc37a2 (isoforms 1&2) and therefore does not differentiate between the two Slc37a2 isoforms. Whereas Slc37a2 isoform2 localizes predominantly to lysosomes, isoform 1 is localized to both lysosomes and the plasma membrane due to its alternative sorting motif found at the extreme C-terminus (SSLALTHPR) (see Figure 2e, d). Therefore, the residual Slc37a2 signal observed in the UB lane is presumably the plasma membrane-bound Slc37a2 isoform 1. Indeed, the UB lane is the peak fraction for the plasma membrane marker V-GLUT1 (see Fig. 1d). Please note that these immunoblot data have since been moved to Supplementary Figure 2a to accommodate the new endogenous Slc37a2 immunostaining panel in intact osteoclasts.

- FP-Slc37a2 is likely to be severely overexpressed under CMV promoter and its co-localization with lysosomal marker could be a result of the overexpression. Also, the “dynamic tubular lysosomal network” could be an artifact of overexpression of FP-Slc37a2. In fact, LysoTracker-positive tubules are more prominent in cells overexpressing FP-Slc37a2 (Fig 2G) as compared to the normal cells (Fig 7B, extended Fig 2A). Immuno-EM with antibodies to the endogenous Slc37a2 should be helpful to resolve this issue.

Response: The existence of the tubular lysosomal network occupied by Slc37a2 has now been comprehensively validated by immunofluorescence confocal microscopy using specific antibodies against endogenous Slc37a2 in osteoclasts in the revised Figure 2a (see also response to Reviewer 2 comment #1). We are therefore confident that our observation is not an artefact of FP-Slc37a2 overexpression. The existence of the tubular lysosomal network is also independently confirmed in live untransfected osteoclasts using the luminal lysosomal tracer LysoTracker (Supplementary Fig. 4b, c) and electron microscopy (Supplementary Fig. 4d). Nonetheless, we agree with Reviewer 2 that FP-Slc37a2 may have the propensity to exaggerate the tubulation phenotype when severely overexpressed in keeping with our observations that Slc37a2 regulates SL resolution and/or tubulation (revised Figure 8). To mitigate this, we exclusively imaged osteoclasts expressing low/moderate levels of FP-Slc37a2.

Is FP-Slc37a2 functional? Could it rescue Slc37a2 KO cells or animals?

Response: Yes, we have confirmed that the FP-Slc37a2 is functional and can rescue the bone resorptive deficit in Slc37a2^{KO} BMM-derived osteoclasts when transduced by lentivirus *in vitro*. Please refer to Figure 7o in our revised dataset.

- In the ovariectomy experiment, the variation in the femoral BV/TV values of KO group seems to be masking a decline in bone mass. However, a statistically significant decline in trabecular thickness is still observed in femoral cancellous bone. It seems these KO mice start with very high bone mass and lose some bone in the femurs (albeit at levels lower compared to controls). It may be more appropriate to call this partial protection in femurs.

Response: This is a fair point. We have toned down our statement accordingly to ‘partial’ protection in femurs.

What happens to cortical thickness in these mice in response to OVX?

Response: This is an excellent question. We have now included detailed assessment of the cortical thickness in response to ovariectomy (OVX) and observed a significant decline in cortical bone thickness in Slc37a2^{KO} (see below and Supplementary Fig. 6b). The exact cellular mechanism underlying cortical bone loss in Slc37a2^{KO} mice is unclear and will require detailed future investigation. Nonetheless, the differences observed between the trabecular and cortical bone parameters in Slc37a2^{KO} OVX mice is in keeping with the wide body of literature documenting distinctions between trabecular and cortical bone loss upon OVX/oestrogen decline that are attributed, in part, to the differential effects of oestrogens, mediated via ER α receptor actions, on distinct hematopoietic and mesenchymal cell types within bone (Manolagas *et al.*, *Nat. Rev. Endocrinol.* 2013; Ponte *et al.*, *Scientific Reports*, 2022).

Is Ctx increased in response to OVX in KO mice?

Response: Unfortunately, we did not sample sufficient bloods from the ovariectomy studies to assess this parameter without a full replication of the experiment which will require a large increase in our available animal numbers and renewal of our ethics approval. This will form part of our ongoing preclinical investigations.

- The presentation of data on the role of *Slc37a2* in the export of monosaccharide sugars from SL (Figure 7D) should be improved. Scale bars and quantification should be included.

Response: As recommended, scale bars have now been added to Figure 7d (now revised Figure 8d) and included quantitation (Figure 8e).

How invertase is getting into secretory lysosomes?

Response: The internalisation and accumulation of invertase into lysosomes is well documented in the literature. Essentially, invertase is endocytosed (via pinocytosis) and trafficked to lysosomes where it hydrolyzes sucrose in the lysosomes (Kao et al., Mol Cell Biochem, 1984, Jadot et al., Eur J Biochem; Ferris et al., J Cell Biol, 1987). This has now been clarified in the text for the reader on page 12 line 389.

- The last sentence of the discussion mentions that KO mice are protected from age-related bone loss, but this data is not included in the manuscript. It would be great to see this data or have this sentence corrected.

Response: Our apologies for the confusion, we have corrected this sentence on page 15, lines 505.

- In the cartoon working model in Fig 8, osteoclasts seem to be producing more SP1 and Tgf β to stimulate the increase in osteoblasts. Increases in mRNA levels of these factors are observed in whole bone preparations of “ femurs” (Fig 5P) but their cellular source is not clear. Tgf β is expressed by numerous cell types in the bone niche including osteoblasts. Moreover, as Tgf β that is released from the bone matrix due to resorption is thought to play a role in coupling resorption to the formation, and the KO mice do not have increased resorption (total ctx is the same in both groups). Therefore, it is unclear how changes in Tgf β mRNA levels fit into the cellular phenotype of these mice. Do the authors measure levels of these factors in osteoclast culture media or their mRNA levels in cultured osteoclasts (similar to what has previously been done with Sp1 of *Ctsk* null osteoclasts)? Are these factors increased due to the increased number of osteoclasts in KOs or is their production/release affected in KO osteoclasts?

Response: This is a good point. We have now measured the mRNA levels of *Sphk1* (SP1) and *TGF- β* in bone marrow monocyte-derived osteoclasts cultured *in vitro* and find that the mRNA expression levels are comparable between WT and *Slc37a2*^{KO} osteoclasts derived from either male and female mice (Supplementary Fig.7d). These data are consistent with our comparative total RNAseq and proteomic analyses (Supplementary Figure 8) where no significant differences in the mRNA and/or protein levels of SP1 and Tgf β were observed between WT and *Slc37a2*^{KO} osteoclasts. Based on these findings, the increased expression of *Sphk1* *in vivo* is presumably a reflection of the increased osteoclast numbers in femurs of *Slc37a2*^{KO} mice rather than increased production/release. On the other hand, considering that TGF- β is expressed in multiple cell types the exact cellular source of the increased *TGF- β* mRNA *in vivo* remains unclear and will require future detailed studies. We have revised the text on page 10, lines 325-329 and our working model (Figure 9) to reflect this.

- The presence of cartilaginous lesions in femurs is very interesting. Are these lesions observed only in femurs or are they present in the vertebral cortex as well?

Response: This is an excellent question. We observe cartilaginous lesions both in the femurs and tibiae but not in the vertebral cortex of *Slc37a2*^{KO} mice. This has now been added to the text page 9, lines 279-280. These data are depicted below and are presented as Supplementary Figure 7a.

Minor comments:

- Please change the % change in cancellous bone mass in line 259 regarding Fig 3E to fold change as opposed to % change.

Response: Thanks. ‘% change’ has been replaced with ‘fold change’ as recommended.

- Please clarify whether the femur gene expression is whole femur or flushed femur etc. in fig 5 E and P.

Response: The femur gene expression is from ‘whole’ femur. This has now been clarified in the text.

- The addition of landmarks or yellow dash lines would help orient readers in the KO image of Fig 4G.

Response: As suggested, the Figure (now revised Figure 5g) has been updated to include a yellow dash line to better orient the reader.

- (i) Evidence that elevated Rankl observed in this model is due to the increased population of osteoblasts as depicted in the cartoon, or (ii) a sentence or two on other possible cellular sources of Rankl in this model would be a nice addition.

Response: Thanks, this is a valuable suggestion. We have updated the figure (Figure 9) to reflect that osteocytes are also a major cellular source of RANKL (*Xiong et al., Nat. Med. 2011; Nakashima T et al., Nat. Med. 2011*).

REVIEWERS' COMMENTS

Reviewer #1 (Remarks to the Author):

Authors addressed my concerns. I have no further comments.

Reviewer #2 (Remarks to the Author):

In the revised manuscript, the authors thoroughly responded to all our comments, and this excellent work is now suitable for publication in Nature Communications.